# WHITTLE INDEX WITH MULTIPLE ACTIONS AND STATE CONSTRAINT FOR INVENTORY MANAGEMENT

**Chuheng Zhang[1], Xiangsen Wang[2], Wei Jiang[3], Xianliang Yang[1],**
**Siwei Wang[1],[*] Lei Song[1], Jiang Bian[1]**
[1]Microsoft Research Asia  [2]Beijing Jiaotong University  [3] University of Illinois Urbana-Champaign
`chuhengzhang@microsoft.com`
`wangxiangsen@bjtu.edu.cn, weij4@illinois.edu`
`{xianliang.yang, siweiwang, lei.song, jiang.bian}@microsoft.com`

## ABSTRACT

Whittle index is a heuristic tool that leads to good performance for the restless bandits problem. In this paper, we extend Whittle index to a new multi-agent reinforcement learning (MARL) setting with multiple discrete actions and a possibly changing constraint on the state space, resulting in WIMS (Whittle Index with Multiple actions and State constraint). This setting is common for inventory management problems, where each agent chooses a replenishing quantity level for the corresponding stock-keeping-unit (SKU) such that the total profit is maximized while the total inventory does not exceed a certain limit. Accordingly, we propose a deep MARL algorithm based on WIMS for inventory management, and evaluate our algorithm empirically on real large-scale inventory management problems with up to 2307 SKUs. The results show that our algorithm outperforms operation-research-based methods and baseline MARL algorithms.

## 1 INTRODUCTION

The inventory management problem is one of the most important problems in operation research (OR) with a history dating back to Fukuda (1964). The objective of the problem is to balance the supply and demand of different stock-keeping-units (SKUs) in the face of stochastic customer demands and vendor lead time (VLT). A good strategy for inventory management can not only reduce the operational cost and increase the profit but also improve customer satisfaction (Eckert, 2007).

Traditional methods from the operation research community solve the problem using dynamic programming (DP). However, these methods often rely on strong assumptions such as *i.i.d.* customer demands and deterministic lead time (Kaplan, 1970; Ehrhardt, 1984) which make them hard to apply for a general setting. While approximate DP methods (Halman et al., 2009; Fang et al., 2013; Chen & Yang, 2019) can solve for more problem formulations by relaxing the assumptions, they still rely heavily on problem-specific expertise or assumptions, e.g., the zero or one period leading time assumption (Halman et al., 2009). In contrast, reinforcement learning (RL) based methods 1) do not rely on strong assumptions and are applicable to general scenarios; 2) can exploit the underlying pattern from real data (instead of solving for fixed distributions of customer demands and VLT); and 3) enable efficient inference instead of, e.g., planning based on the prediction in the inference time as in (Qi et al., 2020). However, it is quite inefficient for RL with a single agent to learn a global policy to make decisions for all SKUs due to the exponentially large state and action space (Jiang & Agarwal, 2018). Therefore, it is natural to adopt the multi-agent reinforcement learning (MARL) paradigm, where each SKU is controlled by one agent whose state and action spaces are localized.

In this paper, we consider a constrained MARL setting for inventory management where a possibly changing constraint on the *total* inventory capacity couples the otherwise *independent* dynamics of a large number of SKUs. We refer to such a structure as *weak coupling* hereinafter. Existing MARL algorithms face some challenges in this setting. The first challenge is the trade-off between efficiency and performance. On the one hand, existing centralized-training-based MARL algorithms (e.g., MADDPG (Lowe et al., 2017), MAPPO (Yu et al., 2021), and COMA (Foerster et al., 2018))

---
[*]Corresponding Author

need to estimate a centralized critic that receives the global state (i.e., the concatenation of local states of all SKUs) as the input. However, in a typical industrial scenario, we have to manage up to thousands of SKUs, which leads to computational inefficiency due to the exponentially large global state space. On the other hand, existing independent-learning-based MARL algorithms (e.g., IQL (Tan, 1993) and IPPO (de Witt et al., 2020)) are efficient but may result in poor performance due to the ignorance of the global constraint. The second challenge is that the limit of the total inventory capacity can change across different seasons (due to promotion, holidays, etc. for inventory management). Accordingly, previous MARL algorithms need to re-train the model on new inventory limits or call for the meta RL formulation (see e.g., Finn et al., 2017; Nichol & Schulman, 2018) which complicates the algorithm/model and results in higher time complexity.

To deal with the above challenges, we resort to Whittle index (Whittle, 1988) proposed as a heuristic but effective tool to solve the restless bandits problem which presents a similar structure. In restless bandits, the dynamics of different arms are independent MDPs (cf. the dynamics of different SKUs are independent in our setting) but only $M$ out of the $N$ arms are allowed to be activated on each time step (cf. the global inventory capacity constraint in our setting), and an algorithm needs to learn how to activate arms to earn reward as much as possible. The basic idea of Whittle index is to estimate the maximum affordable cost for activating each arm in its local state, and then activate the $M$ arms with highest maximum affordable cost. By this way, it decouples the global control into several local learning procedures. However, Whittle index cannot be adopted directly to our setting due to the following two gaps between our setting and restless bandits: 1) The constraint is imposed on the global state (e.g., on the total inventory capacity) in inventory management whereas the constraint is imposed on the global action (i.e., the number of activated arms) in restless bandits; 2) There are multiple actions for each SKU (e.g., multiple candidate replenishment quantities) in inventory management whereas the action for each arm is binary (i.e., activate or not) in restless bandits.

In this paper, we extend Whittle index to bridge the two gaps by defining a new form of Whittle index called WIMS (Whittle Index with Multiple actions and State constraint). WIMS measures the cost of unit budget consumption instead of the cost per arm activation (since our constraint is on the total inventory capacity but not the total replenishment quantity) to bridge the first gap, and generates the critical points of maximum affordable costs for different actions (i.e., a vector) on a given local state instead of only a scalar to bridge the second gap. Then, we propose a practical MARL algorithm called WIMSN that combines WIMS with the neural network to solve real inventory management problems in a computationally efficient way. We evaluate our algorithm based on real data of up to 2307 SKUs, and the experimental results indicate that our algorithm can outperform not only operation-research-based (OR-based) methods but also previous MARL algorithms.

The contributions of our paper are summarized as follows:

- We propose WIMS (Whittle Index with Multiple actions and State constraint) that extends Whittle index to a new constrained MARL setting where a possibly changing constraint is imposed on the global state and there are multiple discrete actions for each agent.
- We propose WIMSN that combines WIMS with neural networks, resulting in a practical algorithm that solves real inventory management problems efficiently.
- We conduct extensive experiments to demonstrate that WIMSN can outperform the OR-based methods as well as previous MARL algorithms.

## 2 RELATED WORK

**Inventory Management.** In the OR community, representative strategies for inventory management include the base-stock policy (Clark & Scarf, 1960; Kaplan, 1970), the $(s, S)$-policy (Ehrhardt, 1984), and the $(R, Q)$-policy Chen (2000), which are proved to be optimal under specific assumptions (see Chen, 2000). However, these assumptions are too strong to hold in practice and researchers start to use the approximate form of dynamic programming (Halman et al., 2009; Fang et al., 2013; Chen & Yang, 2019) instead of the exact form (Goldberg et al., 2016; Huh et al., 2009).

To better exploit the patterns from big data, reinforcement learning methods have been adopted to solve the inventory management problem (see e.g., Giannoccaro & Pontrandolfo, 2002; Jiang & Sheng, 2009; Kara & Dogan, 2018; Barat et al., 2019; Gijsbrechts et al., 2019; Oroojlooyjadid et al., 2017; 2020). While these methods are able to outperform traditional approaches in different scenarios

such as the multi-echelon setting and the setting with volatile customer demands and bullwhip effects, they can hardly scale up to thousands of SKUs efficiently due to the ignorance of the weak coupling structure in the problem. Ding et al. (2022) considers a similar setting to ours and can scale up by decoupling the learning of different SKUs conditioned on the fixed context sequence where the context indicates the allowed capacity for each SKU. However, the model is trained under a given constraint level, and they need to retrain the model when the constraint level changes.

**Whittle Index.** Whittle index (Whittle, 1988) is a heuristic tool that is effective in solving the restless bandits problem (a PSPACE-hard problem) (Papadimitriou & Tsitsiklis, 1994). Restless bandits problem generalizes many real-world problems including HVAC control (Wei et al., 2017), wireless sensor scheduling in cyber-physical systems (Leong et al., 2020), and dynamic multi-channel access in wireless networks (Wang et al., 2018),(Wang et al., 2020), and Whittle index is shown to be effective in these real-world applications (see e.g., Le Ny et al., 2008; Li et al., 2022). Some existing researches also try to extend Whittle index to restless bandits with multiple actions (Killian et al., 2021a; Xiong et al., 2022; Hodge & Glazebrook, 2015; Killian et al., 2021b), but they are still dealing with constraints on the action space and cannot be applied to our setting (where the constraint is on the state space). In this paper, we further extend Whittle index to a new MARL setting where a constraint on the state space couples the otherwise independent agents (i.e., the weak coupling structure), which could be applied in a wide range of applications, e.g., inventory management (Ding et al., 2022), portfolio management in quantitative finance (Jiang et al., 2017), queue scheduling (Chen et al., 2021), recommending systems (Liao et al., 2022), and the air campaign planning task (Meuleau et al., 1998).

## 3 PRELIMINARY

**Notations.** $\delta_a$ is used to represent Dirac delta distribution centered at $a$. $[N]$ is a shorthand for $\{1, 2, \cdots, N\}$. $\mathbb{I}\{\text{cond}\}$ is the identity function which equals 1 if cond = True and equals 0 otherwise. Given a vector $x$, we use $[x]_i$ to represent the $i$-th component of this vector.

**Restless Bandits.** The restless bandits problem is defined as follows: in each round $t$, a control policy observes the state of each of the $N$ arms (denoted by $s_t^i$ for all $i \in [N]$) and activates $M$ arms (denoted by $a_t^i \in \{0, 1\}$ where 1 represents activation). The joint state space is the Cartesian products of individual spaces, i.e., $\bar{\mathcal{S}} := \mathcal{S}^N$, and the joint action space is defined as $\bar{\mathcal{A}} := \{a \in \mathcal{A}^N | \sum_{i \in [N]} a_i = M\}$ (here $\mathcal{A} := \{0, 1\}$). Then, the policy receives a reward $r_t^i \sim R^i(s_t^i, a_t^i)$ on every arm $i$, with $R^i(s_t^i, 0) = \delta_0$ (i.e., if we do not activate arm $i$ at this time step, its reward must be 0), and the state of each arm transits to the next state $s_{t+1}^i \sim P^i(\cdot|s_t^i, a_t^i)$. The objective is to find a policy $\bar{\pi} : \bar{\mathcal{S}} \to \Delta(\bar{\mathcal{A}})$ that maximizes $\mathbb{E}[\sum_{t=0}^{\infty} \sum_{i=1}^{N} \gamma^t r_t^i | \bar{\pi}]$ where $\gamma$ is the discount factor.

**Whittle Index Policy.** The Whittle index policy first decouples the joint optimization problem of $N$ arms into individual solutions for each arm, and then use a simple way to combine these solutions.

Therefore, we can first only consider an arbitrary arm $i \in [N]$. We suppose the agent has to pay a cost $\lambda$ every time it activates this arm, and the objective is to find a policy $\pi \in \Pi : \mathcal{S} \to \Delta(\mathcal{A})$ (i.e., determine whether to activate this arm after observing the current state $s_t^i$) to maximize $\mathbb{E}[\sum_{t=0}^{\infty} \gamma^t (r_t^i - \lambda a_t^i)|\pi]$. Accordingly, we can define the Q function and the optimal Q function as $Q_\lambda^\pi(s, a) := \mathbb{E}\left[\sum_{t=0}^{\infty} \gamma^t (r_t^i - \lambda a_t^i)|s_0^i = s, a_0^i = a, \pi\right]$ and $Q_\lambda^*(s, a) = \max_\pi Q_\lambda^\pi(s, a)$ respectively. The corresponding optimal policy can be defined as $\pi_\lambda^* \in \arg\max_\pi Q_\lambda^\pi(s, a)$.

Intuitively, the higher the cost $\lambda$, the less likely the optimal activation policy $\pi_\lambda^*$ would activate the arm in a given state. This intuition can be formulated as the following *indexability* property:

**Definition 3.1** (Indexability). *Let $\mathbb{S}(\lambda)$ be the set of the states where the optimal policy $\pi_\lambda^*$ activates this arm under activation cost $\lambda$. Then an arm is indexable if $\mathbb{S}(\lambda)$ decreases monotonically from $\mathcal{S}$ to $\emptyset$ when $\lambda$ increases from $-\infty$ to $+\infty$.*

The Whittle index $f(s)$ for a given state $s \in \mathcal{S}$ is defined as the maximum activation cost $\lambda$ the optimal policy $\pi_\lambda^*$ is willing to pay to activate the arm on state $s$, i.e.,

**Definition 3.2** (Whittle Index). *If an arm is indexable, then its Whittle index of each state $s$ is defined as $f(s) := \sup_\lambda\{\lambda : s \in \mathbb{S}(\lambda)\}$.*

The Whittle index can be determined by estimating Q values owing to the following property:

**Proposition 3.3** (Learnability). *If an arm is indexable, then: 1) when $\lambda < f(s)$, $Q_\lambda^*(s,1) > Q_\lambda^*(s,0)$; 2) when $\lambda > f(s)$, $Q_\lambda^*(s,1) < Q_\lambda^*(s,0)$; 3) when $\lambda = f(s)$, $Q_\lambda^*(s,1) = Q_\lambda^*(s,0)$.*

Given a global state $s = (s_1, s_2, \cdots, s_N)$, the Whittle index policy will first find a $\lambda_g$ such that $\sum_{i=1}^N \mathbb{I}\{f(s_i) \geq \lambda_g\} = M$ and then activate the arms with $f(s_i) \geq \lambda_g$.

# 4 INVENTORY MANAGEMENT AND WIMS

In this section, we will first introduce the multi-agent reinforcement learning (MARL) formulation of the inventory management problem. Then, we will adapt the Whittle index policy to our setting by bridging the gaps between restless bandits and the MARL formulation. Specifically, we formulate the definition of Whittle Index for Multiple actions and State constraint (WIMS), and obtain some of its properties. At last, we calculate WIMS and the corresponding policy on a simplified inventory management task for illustration.

## 4.1 PROBLEM FORMULATION

In this paper, we formulate the inventory management problem considered as follows: There are totally $N$ agents, and each agent represents a single SKU. At the $t$-th time step, the policy receives the local states of $N$ agents $\{s_t^i\}_{i=1}^N$, while local state $s_t^i$ contains the current features of the $i$-th SKU, e.g., the resource occupation, the procurement/selling prices, the historical demands, the inventory/in-transit levels, etc. Then, the policy needs to decide the local actions $\{a_t^i\}_{i=1}^N$ for all the $N$ SKUs, i.e., the indices of a list of candidate quantities for replenishment orders. We denote the local state and action space of each agent as $\mathcal{S}$ and $\mathcal{A} := [A]$ respectively, and denote the joint state as $\bar{s}_t = (s_t^1, \cdots, s_t^N) \in \bar{\mathcal{S}}$ and the joint action as $\bar{a}_t = (a_t^1, \cdots, a_t^N) \in \bar{\mathcal{A}} = \mathcal{A}^N$. Then, the environment returns a reward $r_t \sim R(\bar{s}_t, \bar{a}_t)$ and transits to the next state $\bar{s}_{t+1} \sim P(\cdot|\bar{s}_t, \bar{a}_t)$.

The $N$ agents are coupled weakly only through a global constraint $\sum_{i=1}^N l(s_t^i) \leq L_t$, where $l(s_t^i)$ is the resource occupation of the $i$-th agent (or the inventory of the $i$-th SKU) at the $t$-th time step, and $L_t$ is the budget level at the $t$-th time step. In other words, without this constraint, we can decompose the reward and the transition dynamics as $R(s_t, a_t) = \sum_{i=1}^N R^{\text{dec}}(s_t^i, a_t^i)$ and $P(\cdot|s_t, a_t) = \prod_{i=1}^N P^{\text{dec}}(\cdot|s_t^i, a_t^i)$ for some decomposed reward $R^{\text{dec}}$ and transition dynamics $P^{\text{dec}}$.

## 4.2 WHITTLE INDEX WITH MULTIPLE ACTIONS AND STATE CONSTRAINT (WIMS)

In this subsection, we only consider one of the agents in the multi-agent system and omit the superscript $i \in [N]$ for simplicity.

We suppose that there is a cost $\lambda$ for the occupation of each unit of shared resource (in comparison, the Whittle index policy uses $\lambda$ to represent the activation cost of each arm). The Q function for a given policy $\pi$ and the optimal Q function are defined as

$$Q_\lambda^\pi(s,a) = \mathbb{E}\left[\sum_{t=0}^\infty \gamma^t (r_t - \lambda l(s_t)) \Big| s_0 = s, a_0 = a, \pi\right], \qquad Q_\lambda^*(s,a) = \max_\pi Q_\lambda^\pi(s,a), \quad (1)$$

for all $s \in \mathcal{S}$, $a \in \mathcal{A}$ where the expectation is taken over all possible trajectories collected by rolling out the policy $\pi$. According to Puterman (1990), there exists a deterministic optimal policy for any given MDP, and we denote this deterministic policy as $\pi_\lambda^* : \mathcal{S} \to \mathcal{A}$, i.e., $\pi_\lambda^*(s) \in \arg\max_\pi Q_\lambda^\pi(s,a)$.

Similarly, we can define the indexability condition for our setting.

**Definition 4.1** (Indexability). *For all $\alpha \in [A-1]$, let $\mathbb{S}_\alpha(\lambda)$ be the set of the states where the optimal policy chooses action $a > \alpha$ under unit inventory cost $\lambda$. Then an agent in the multi-agent system formulated in Section 4.1 is indexable if for any $\alpha$, $\mathbb{S}_\alpha(\lambda)$ decreases monotonically from $\mathcal{S}$ to $\emptyset$ when $\lambda$ increases from $-\infty$ to $+\infty$.*

**Lemma 4.2** (Sufficient condition of indexability). *An agent in the multi-agent system is indexable if it satisfies the following action monotonicity condition: for any local state $s \in \mathcal{S}$ and two costs $\lambda_1 > \lambda_2$, the optimal policies $\pi_{\lambda_1}^*, \pi_{\lambda_2}^*$ of this agent satisfy $\pi_{\lambda_1}^*(s) \leq \pi_{\lambda_2}^*(s)$.*

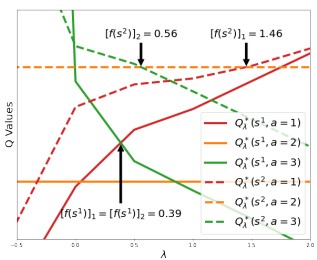 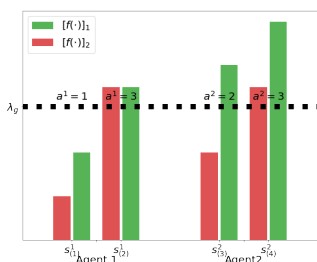 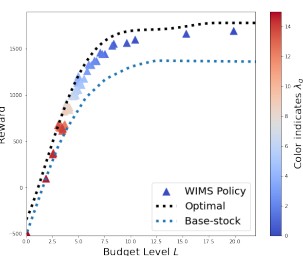

Figure 1: (a) Illustration on how WIMS is calculated based on the Q values learned in the simplified inventory management task. We show the case on state $s^i = (i, 0, 8)$ and the Q values are normalized by subtracting $Q^*_\lambda(s^i, a = 2)$ from $Q^*_\lambda(s^i, \cdot)$ where $a = 1, 2, 3$ indicates ordering $0, 1, 2$ units for replenishment respectively and $\lambda$ represents the unit inventory cost. (b) Illustration on how to select actions based on WIMS and a given $\lambda_g$. The bars indicate the WIMS of two agents on different local states. The red bars indicate the values of $[f(s)]_2$ and the green bars indicate the values of $[f(s)]_1$. (c) The WIMS policy combined with different $\lambda_g$ results in near-optimal policies under different budget levels. Each triangle represents the performance of the WIMS policies with different $\lambda_g$. The black and blue dotted lines represent the best attainable reward under different inventory levels and the performance of the best base-stock policies.

Lemma 4.2 indicates that the above action monotonicity condition can lead to indexability naturally, and its proof is deferred to Appendix B.1. We provide intuition on why the condition is natural for inventory management in Appendix B.2. Roughly speaking, the condition requires that an optimal policy should not reduce the replenishment quantity when the inventory cost $\lambda$ decreases.

Since we have $A$ candidate actions, we define the Whittle index as a vector, i.e., $f(s) = (\lambda_1, \cdots, \lambda_{A-1})$, where the $\alpha$-th element represents the maximum inventory cost the optimal policy is willing to pay to choose an action from $\{a | a > \alpha\} \subset \mathcal{A}$. Its formal definition is given below.

**Definition 4.3** (WIMS). *If an agent is indexable, then the WIMS of each state $s$ is a vector $f(s)$ such that its $\alpha$-th term is $[f(s)]_\alpha := \sup_\lambda \{\lambda : s \in \mathbb{S}_\alpha(\lambda)\}, \forall \alpha \in [A-1]$.*

By observing $\mathbb{S}_\alpha(\lambda) \subseteq \mathbb{S}_{\alpha'}(\lambda)$ if $\alpha' < \alpha$, we can obtain the following proposition:

**Proposition 4.4.** $[f(s)]_1 \geq [f(s)]_2 \geq \cdots \geq [f(s)]_{A-1}$ *for any state $s$.*

The WIMS values can be determined by comparing the following two types of aggregated Q values:

$$Q^*_\lambda(s, \leq j) = \max_{a \leq j} Q^*_\lambda(s, a), \qquad Q^*_\lambda(s, > k) = \max_{a > k} Q^*_\lambda(s, a). \qquad (2)$$

**Lemma 4.5** (Learnability of WIMS). *If an agent in the multi-agent system is indexable, for any $s \in \mathcal{S}$ and $\alpha \in [A-1]$, we have that: 1) when $\lambda < [f(s)]_\alpha$, $Q^*_\lambda(s, > \alpha) \geq Q^*_\lambda(s, \leq \alpha)$; 2) when $\lambda > [f(s)]_\alpha$, $Q^*_\lambda(s, > \alpha) \leq Q^*_\lambda(s, \leq \alpha)$; 3) when $\lambda = [f(s)]_\alpha$, $Q^*_\lambda(s, > \alpha) = Q^*_\lambda(s, \leq \alpha)$.*

This lemma leads to the update rule that we later use to determine WIMS in Section 5.

Given a global state $\bar{s}_t = (s^1_t, s^2_t, \cdots, s^N_t)$ and a global cost $\lambda_g$, the WIMS policy takes the action

$$a^i_t = \sum_{\alpha < A} \mathbb{I}\{[f(s^i_t)]_\alpha \geq \lambda_g\} + 1 \qquad (3)$$

for each agent indexed by $i$, i.e., selecting the maximum action $\alpha$ whose corresponding affordable cost ($[f(s^i)]_\alpha$) exceeds $\lambda_g$. The global cost $\lambda_g$ is determined by the constraint level $L_t$ and $\lambda_g$ can be updated dynamically in an online manner: we decrease $\lambda_g$ when the constraint is not violate and otherwise increase it. More details will be introduced in Section 5.

**Lemma 4.6** (Asymptotic optimality of the WIMS policy). *When $L_t \to \infty$, the WIMS policy with the global cost $\lambda_g = 0$ is optimal.*

The proofs of Lemma 4.5 and 4.6 are deferred to Appendix B.3 and B.4 respectively.

### 4.3 WIMS FOR INVENTORY MANAGEMENT: A TOY EXAMPLE

To justify the formulation of WIMS, we design a tabular inventory management task to illustrate how the policies based on WIMS can be used to solve the inventory management problem and indicate the optimality gap of these policies for inventory management.

We construct a minimum tabular example on $N = 2$ SKUs with stochastic demand and stochastic VLT as follows: The state of the $i$-th SKU $s^i = (i, I^i, T^i)$ consists of the index and two integers representing the in-stock quantity and the in-transit quantity respectively. We incorporate the index into the state features to allow for the reward/dynamics of the two SKUs to be different. For compact state representation, we clip the integers to $[0, 9]$ and thus there are $S = 200$ possible local states. The action space $\mathcal{A} = [3]$ with the list of candidate replenishing quantities to be $\{0, 1, 2\}$. We set the unit profit, the order cost, the unit backlog cost to $p = (10, 20)$, $o = (10, 10)$, and $b = (3, 3)$ respectively where the two numbers stand for the values for two SKUs. We consider a constraint on the total inventory capacity, i.e., $\sum_{i=1}^2 l(s^i) \leq L$ with $l(s^i) = I^i$ and $L$ to be constant. The demand on each day is sampled uniformly from $\{0, 1, 2\}$ and each unit of in-transit product arrives w.p. $1/3$ on each day. We defer the details of the inventory management task in Appendix A.

We solve this simplified task by first learn $Q^*_\lambda$ for different discretized $\lambda$s using Q-learning and calculate the WIMS indices following Definition 4.3. Then, by using different $\lambda_g$, we can obtain a set of WIMS policies which will be shown to be near optimal for different budget levels.

**How to calculate WIMS?** To illustrate this, we fix a local state to see how WIMS is calculated based on the learned Q values. We fix the local state $s^i = (i, 0, 8)$ for the two SKUs and show the Q values under different $\lambda$s in Figure 1(a). Owing to the action monotonicity condition, the optimal action on a given state decreases monotonically as $\lambda$ increases. Therefore, when $\lambda$ changes from $-\infty$ to $+\infty$, the maximum Q value changes from the Q value of the largest action (cf. the green lines) to that of the smallest action (cf., the red lines) monotonically. This ensures that there are at most $A - 1$ critical points which are defined as WIMS. Note that it is possible that some action is never optimal for any $\lambda$. For example, for the first SKU, the optimal action is $a = 3$ (green solid) when the unit inventory cost $\lambda < 0.39$; whereas the optimal action becomes $a = 1$ (red solid) when the unit inventory cost $\lambda > 0.39$. In this case, the two dimensions of WIMS on this state for the first SKU are both 0.39 (this indicates that $a = 2$ is never optimal on this state). For the second SKU, when the unit inventory cost $\lambda$ increases, the optimal action transits $a = 3 \to 2 \to 1$ and $[f(s^2)]_2 = 0.56, [f(s^2)]_1 = 1.46$ are the two critical points for the transition. Intuitively, WIMS can be interpreted as the priority (or the maximum affordable cost) assigned to each local state for ordering at a larger quantity for that SKU. Since the second SKU is more profitable than the first SKU, it is reasonable to see that the WIMS for the second SKU is larger than that of the first SKU.

**How to implement the WIMS policy?** When a global cost $\lambda_g$ is given, agent $i$ selects action $\sum_{\alpha < A} \mathbb{I}\{f(s^i_t)_\alpha \geq \lambda_g\} + 1$, which is the same as the largest action $a + 1$ as long as the corresponding WIMS $[f(s^i)]_a$ on the local state $s^i$ (i.e., the maximum affordable cost) is larger than the given $\lambda_g$. By using different $\lambda_g$ the policy can trade off between the cumulative reward and the constraint level. We illustrate this process in Figure 1(b) where we consider two states for each of the two agents and three actions. The three actions can be selected based on relationship between the WIMS vector of the local state and the given $\lambda_g$. For example, when $[f(s^1_{(1)})]_2 < [f(s^1_{(1)})]_1 < \lambda_g$ we choose $a^1 = 0 + 1 = 1$; and when $[f(s^2_{(3)})]_2 < \lambda_g < [f(s^2_{(3)})]_1$ we choose $a^2 = 1 + 1 = 2$.

**The optimality of the WIMS policy.** To estimate the optimality gap of the WIMS policy, we compare it with the optimal policies and the best base-stock policies (Clark & Scarf, 1960). The optimal policies under different inventory levels are obtained by searching using the genetic algorithm from pymoo (Blank & Deb, 2020). Note that, we search for the *global* optimal policy for each budget level, whereas our WIMS policy only needs to learn for *local* agents and obtains the indices that work for different inventory cost levels. For comparison, we also evaluate the performance of base-stock policies which is a popular method from operation research. Given a base-stock level $b^i$ for each agent, the policy orders $\max(b^i - I^i - T^i, 0)$ units for the agent in state $s^i$. We iterate over all possible base-stock levels $(b^1, b^2)$ and record the best performance of these base-stock policies under different budget levels (i.e., the Pareto frontier). We present the results in Figure 1(c). We can see that, while having the advantages of computational efficiency, our algorithm outperforms the best base-stock policies and performs on par with the globally optimal policies.

**Discussion.** One advantage of the WIMS policy over other MARL algorithms is that it only needs to learn the indices once for each SKU to obtain good policies over different constraint levels. In contrast, to adopt popular MARL algorithms (such as IQL (Tan, 1993)), we first need to transform the hard constraint into penalty terms in the reward function and re-train the model when the constraint

---

**Algorithm 1** WIMSN-c: Learn the WIMS policy with neural network on **c**ontinuous $\lambda$

---

1: **Initialize:** The Q network $Q_\theta : \mathcal{S} \times \mathcal{A} \times \mathbb{R} \to \mathbb{R}$ and the Whittle index network $f_\phi : \mathcal{S} \to \mathbb{R}^{A-1}$
2: ▷ *Phase 1: Train agents independently without any constraint*
3: Initialize the replay buffer $\mathcal{D}$ with warming-up samples collected by the random policy
4: **for** each iteration **do**
5:     Select $s \sim \mathcal{D}$ and $\alpha \in [A-1]$ uniformly at random and obtain $\lambda \leftarrow [f_\phi(s)]_\alpha$
6:     Roll out $\epsilon$-greedy policy of $\mu_\lambda(\cdot) := \arg\max_a Q_\theta(\cdot, a, \lambda)$ and store new transitions to $\mathcal{D}$
7:     Sample a batch $\mathcal{B} := \{(s, a, r, s')\}$ from $\mathcal{D}$
8:     Update the Q-network and the Whittle index network with (4) and (5) respectively on $\mathcal{B}$
9: **end for**
10: ▷ *Phase 2: Execute agents jointly with possibly changing global constraint*
11: Initialize $\lambda_g$ with a reasonably good prior value
12: **for** each step **do**
13:     Take $a^i = \sum_{\alpha < A} \mathbb{I}\{[f_\phi(s^i)]_\alpha \geq \lambda_g\} + 1$ for each agent $i \in [N]$
14:     Update $\lambda_g$ following (6)
15: **end for**

---

level changes. For this reason, we do not compare WIMS with MARL baselines in this simplified task and defer the comparison to the experiments on real inventory management tasks in Section 6.

## 5 PRACTICAL ALGORITHM

In this section, we will introduce a practical algorithm called WIMSN (as shown in Algorithm 1) that learns the WIMS policy using neural networks. In the training phase, we train two networks: the Q network $Q_\theta : \mathcal{S} \times \mathcal{A} \times \mathbb{R} \to \mathbb{R}$ that estimates the Q value given local state $s$, action $a$, and unit inventory cost $\lambda$, and the Whittle index network $f_\phi : \mathcal{S} \to \mathbb{R}^{A-1}$ that outputs the Whittle index vector for each local state $s$. (With slight abuse of notation, we move the subscript $\lambda$ to the input variable of Q in this section.) In the execution phase, we maintain a dynamically updated global cost $\lambda_g$, which is used for making decisions combined with $f_\phi$.

**Training phase.** In the training phase, each agent can learn independently in an environment without the global constraint. This makes our algorithm scalable to a large number of agents. In our case, we learn the Q network and the Whittle index network whose parameters are shared across different agents. The network is trained through iterations, each of which consists of two steps: the sample collection step (Line 5-6) and the network update step (Line 7-8).

In the sample collection step, we first select a random cost $\lambda$ as one of the Whittle indices of a random state from the replay buffer. Then, we collect samples by taking the action that yields the maximum Q value conditioned on $\lambda$, and store the transitions to the replay buffer $\mathcal{D}$.

In the network update step, we sample a batch of transitions $\mathcal{B}$ from the replay buffer $\mathcal{D}$ and update the networks based on $\mathcal{B}$. The Q network is updated with the following gradient update rule:

$$\theta \leftarrow \theta - \beta_1 \nabla_\theta \mathbb{E}_{(s,a,r,s') \sim \mathcal{B}} \mathbb{E}_\lambda \Big[ Q_\theta(s, a, \lambda) - \Big( r - \lambda l(s_t) + \gamma \max_{a'} Q_\theta(s', a', \lambda) \Big) \Big]^2, \quad (4)$$

where $\beta_1$ is the learning rate, and the expectation $\mathbb{E}_\lambda$ is over different $\lambda = [f(s)]_\alpha$ for random $s \in \mathcal{D}$ and $\alpha \in [A-1]$. The Whittle index network is updated with the following gradient update rule:

$$\phi \leftarrow \phi + \beta_2 \mathbb{E}_{(s,a,r,s') \sim \mathcal{B}} \Big[ \sum_{\alpha=1}^{A-1} \nabla_\phi [f_\phi(s)]_\alpha \cdot \big( Q_\theta(s, > \alpha, [f_\phi(s)]_\alpha) - Q_\theta(s, \leq \alpha, [f_\phi(s)]_\alpha) \big) \Big], \quad (5)$$

where $Q_\theta(s, > \alpha, \lambda) := \max_{a > \alpha} Q_\theta(s, a, \lambda)$, and $Q_\theta(s, \leq \alpha, \lambda) := \max_{a \leq \alpha} Q_\theta(s, a, \lambda)$. $\beta_2$ is the learning rate and we use $\beta_2 \ll \beta_1$ as updating $f_\phi$ relies on well-estimated $Q_\theta$ (Robledo et al., 2022).

*Remark.* The Q network $Q_\theta$ and the Whittle index network $f_\phi$ are coupled through simultaneous training: On one hand, $Q_\theta$ is trained on the $\lambda$s sampled from $f_\phi$ to gradually estimate more accurate Q values around these $\lambda$s. On the other hand, with a more accurate estimation of the Q values around the current Whittle indices, $f_\phi$ can be updated more precisely.

**Execution phase.** In the execution phase, we execute the policy determined by a global cost $\lambda_g$. Specifically, given the global cost $\lambda_g$, we select the action for each agent based on Eq. (3). We also refer the readers to Figure 1(b) for an illustration on this process.

The global cost $\lambda_g$ is updated dynamically. The update rule is as follows:

$$\lambda_g^{(t+1)} \leftarrow \lambda_g^{(t)} + \beta_3(\hat{L}^{(t)} - L_t), \text{ with } \hat{L}^{(t)} := \max_{t-T < \tau \leq t} \sum_{i=1}^{N} l(s_\tau^i) \tag{6}$$

where $\lambda_g^{(0)}$ is the prior value, $\hat{L}^{(t)}$ is the maximum global resource occupation over a window $T$, and $\beta_3$ is the learning rate. When the budget is not fully utilized (i.e., $\hat{L}^{(t)} - L_t < 0$), we decrease the global cost such that the agents are more willing to take aggressive actions (e.g., order more quantities for replenishment in inventory management). Otherwise, if the constraint is violated, we will increase the global cost so that the agents will not order too many quantities for replenishment afterwards.

## 6 EXPERIMENTS

In this section, we design experiments to 1) compare the performance of WIMSN with other baseline MARL algorithms and OR methods with fixed budget levels; and 2) evaluate how WIMSN adapts when the budget level changes. We provide the implementation details in Appendix C.

**Settings.** We evaluate WIMSN in several inventory management tasks with different scales (i.e., different number of SKUs) and different budget levels (by changing the inventory capacity limit). Our experiment is conducted on a simulator that can simulate the replenishment procedure for multiple SKUs in one store. Instead of sampling demands from some hypothetical distributions, we replay the demand series from an authentic dataset from a retail partner that contains the sales history of four years (2018-2021) with 2307 SKUs. We hold out the data of the last 200 days from the training set and split these data into evaluation set (for parameter tuning) and testing set (for reporting the performance) respectively. In the test procedure, when the total in-stock quantity exceeds the inventory capacity, we reduce equal amount of in-stock quantities of different SKUs such that the total in-stock quantity equals the capacity limit. The dropped quantities do not contribute any reward (neither income nor penalty), just as they have never been replenished. See Appendix A for details.

**OR-Based Methods.** We compare our WIMSN policy with two popular operation research (OR) methods: the base-stock policy (Clark & Scarf, 1960; Kaplan, 1970) and the $(s, S)$-policy (Ehrhardt, 1984). In these methods, we need to specify few parameters to control inventory replenishment. In the baseline version, we find the optimal parameters for each SKU on the training set. In the skyline version, we use those parameters that are optimal *ad-hoc on the testing set*. See Appendix C.1 for the notion of optimality and implementation details.

**Baseline MARL Algorithms.** To apply off-the-shelf MARL algorithms for our constrained MARL setting, we add penalty to reward when the constraint is violated during the training (see also Appendix C.2). We choose IPPO (de Witt et al., 2020) and independent Q-learning (Tan, 1993) with neural networks (Mnih et al., 2013) (IDQN) as the baselines. We do not compare with other MARL algorithms (e.g., MAPPO (Yu et al., 2021)) that require to learn a joint critic for all agents since they are not computationally efficient in our environment with thousands of agents. In addition, we compare with Chen et al. (2021) which updates a global inventory cost for the constrained MARL problem for each budget level. Note that all these MARL baselines need to re-train the model when the budget level or the set of SKUs changes, but WIMS uses the same model for all tasks.

**Versions of WIMSN.** WIMSN-c intertwines the learning of the two networks (i.e., the Q network is trained on different values of continuous $\lambda$ sampled from the output of the Whittle index network). It is also possible to decouple them by only training the two networks on a grid of discretized $\lambda$s, resulting in WIMSN-d with details introduced in Appendix C.4.

**Experiment Results.** We evaluate the above methods on inventory management tasks with 100, 1000, and 2307 SKUs and different capacity limits, and present the results in Table 1. Here are some of our observations. First, IPPO and IDQN tend to violate the constraint limit when the budget is tight[1]. In contrast, WIMSN incurs less constraint violation ($< 1\%$ for all tasks) owing to our constrained optimization formulation. Second, WIMSN earns more profit with less constraint violation than previous methods, especially with a large number of SKUs. This indicates the effectiveness of WIMSN in coordinating a large number of agents. Third, those baselines with a higher profit than

---

[1]Indeed, we can further increase the constraint violation penalty in the reward function to reduce constraint violation for these MARL algorithms, but this causes instability during the training.

| | OR-Based Methods | | | | | | | | RL-Based Methods | | | | | | | | | |
| | Baseline BS | | Baseline SS | | Skyline BS | | Skyline SS | | IPPO | | IDQN | | Primal-Dual | | WIMSN-d | | WIMSN-c | |
| | profit | VIO | profit | VIO | profit | VIO | profit | VIO | profit | VIO | profit | VIO | profit | VIO | profit | VIO | profit | VIO |
|---|---|---|---|---|---|---|---|---|---|---|---|---|---|---|---|---|---|---|
| N100-C2.5k | 7.52 | 20.88% | 8.93 | 16.68% | 10.25 | 6.03% | 10.29 | 7.10% | 6.44 | 38.21% | 5.94 | 34.79% | 5.86 | 8.75% | 9.10 | 0.21% | 8.84 | 0.76% |
| N100-C5k | 11.38 | 6.45% | 12.91 | 4.78% | 14.16 | 0.00% | 13.91 | 0.00% | 13.53 | 28.43% | 12.67 | 24.07% | 11.60 | 0.62% | 12.61 | 0.06% | 12.38 | 0.06% |
| N100-C10k | 15.29 | 2.36% | 15.72 | 1.20% | 16.57 | 0.00% | 16.70 | 0.00% | 18.31 | 16.54% | 16.72 | 16.91% | 14.84 | 0.00% | 16.93 | 0.00% | 17.43 | 0.00% |
| N100-C20k | 16.61 | 0.06% | 17.13 | 0.00% | 17.23 | 0.00% | 17.81 | 0.00% | 17.89 | 7.18% | 16.95 | 8.42% | 16.60 | 0.00% | 18.01 | 0.00% | 17.51 | 0.00% |
| N1k-C25k | 10.54 | 6.55% | 10.93 | 4.86% | 11.80 | 0.00% | 11.83 | 0.00% | 9.55 | 26.38% | 9.13 | 19.67% | 8.65 | 0.74% | 10.83 | 0.15% | 12.05 | 0.47% |
| N1k-C50k | 13.44 | 2.61% | 13.95 | 1.21% | 14.10 | 0.00% | 14.37 | 0.00% | 11.64 | 8.97% | 11.37 | 9.81% | 10.91 | 0.00% | 13.27 | 0.00% | 14.08 | 0.00% |
| N1k-C100k | 14.88 | 0.00% | 15.63 | 0.00% | 15.60 | 0.00% | 16.01 | 0.00% | 16.05 | 4.50% | 17.39 | 5.32% | 14.90 | 0.00% | 16.94 | 0.00% | 17.04 | 0.00% |
| N1k-C200k | 15.90 | 0.00% | 16.48 | 0.00% | 15.96 | 0.00% | 16.51 | 0.00% | 16.54 | 0.88% | 16.24 | 0.00% | 15.87 | 0.00% | 16.93 | 0.00% | 16.26 | 0.00% |
| N2k-C58k | 8.95 | 5.48% | 9.70 | 4.63% | 9.42 | 0.00% | 9.57 | 0.00% | 5.45 | 5.49% | 6.82 | 7.27% | 8.86 | 0.00% | 9.17 | 0.00% | 9.50 | 0.00% |
| N2k-C115k | 11.30 | 0.87% | 11.85 | 0.26% | 11.94 | 0.00% | 12.19 | 0.00% | 9.36 | 0.88% | 9.18 | 0.00% | 11.11 | 0.00% | 11.60 | 0.00% | 12.20 | 0.00% |
| N2k-C231k | 12.85 | 0.00% | 12.99 | 0.00% | 12.94 | 0.00% | 13.42 | 0.00% | 12.02 | 0.00% | 12.38 | 0.00% | 12.67 | 0.00% | 12.58 | 0.00% | 13.14 | 0.00% |
| N2k-C461k | 13.02 | 0.00% | 13.62 | 0.00% | 13.13 | 0.00% | 13.56 | 0.00% | 13.09 | 0.00% | 13.58 | 0.00% | 13.19 | 0.00% | 13.30 | 0.00% | 14.27 | 0.00% |

Table 1: The daily profit per SKU in $\times 10^3$ USD and the ratio of violation on the capacity limit of different algorithms in different inventory management tasks on held-out testing data. For the tasks, "N" indicates the number of SKUs to control and "C" indicates the capacity limit on the shared inventory (i.e., the budget level $L$). We compare WIMSN with OR-based methods and RL-based methods: BS (base-stock policy), SS ($(s, S)$-policy), IPPO (de Witt et al., 2020), IDQN (Tan, 1993), Primal-Dual (Chen et al., 2021), WIMSN-d (with discrete $\lambda$s), and WIMSN-c (with continuous $\lambda$s). See details in Section 6. WIMSN models use the same model for all tasks with the different "N" and "C" while others train separate models for different tasks. We present the average over 5 seeds. We underline Parato optimal solutions and color Parato optimal solutions with constraint violation $< 1\%$ (excluding skyline BS and skyline SS since they are ad-hoc solutions).

WIMSN always incur a higher constraint violation, which means they may behave worse in reality. (In our experiment, constraint violation does not incur any loss, but in reality, a higher constraint violation may lead to additional loss.) Besides, unlike the other methods, WIMSN does not need to re-train the model for different tasks, therefore greatly reducing the computational costs (due to space limit, the computational costs of different methods and the variance across different seeds are provided in Appendix D).

**Online Adaptation for Capacity Limit.** To evaluate the adaptability of WIMSN to the changing budget level, we select a 100-day period to control 1000 SKUs and change the capacity from 50k to 100k on the 50th day. We present the cumulative profit per SKU and the maximum constraint violation over this period in Figure 2. We compare WIMSN-c with two IPPO models trained under the capacity limit 100k and 50k respectively (i.e., C100k/C50k). We observe that the IPPO (C100k) can overflow in the first 50 days and perform poorly due to the inconsistency between capacity limits used in training (100k) and testing (50k), whereas the IPPO (C50k) does not perform good enough for the last 50 days due to overly conservative replenishment strategy. In contract, WIMSN-c performs well across all the time period by adjusting $\lambda_g$ adaptively according to the current capacity limit (cf. the purple curves in

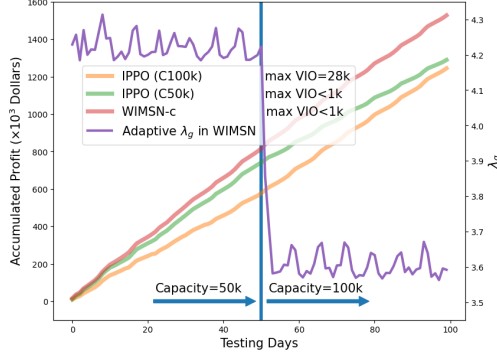

Figure 2: On the task where the capacity limit changes from 50k to 100k on the 50th day, WIMSN-c earns 22.7% and 18.4% more cumulative profit than IPPO (C100k) and IPPO (C50k) respectively while incurring significantly less constraint violation than IPPO (C100k).

Figure 2). When the capacity is tight, $\lambda_g$ fluctuates around 4.2; and when the capacity doubles, $\lambda_g$ adapts quickly and decreases to values around 3.6 to allow agents to take more aggressive actions.

## 7 CONCLUSION

We proposed WIMS (Whittle Index with Multiple actions and State constraint), a novel form of Whittle Index that extends to the multi-agent reinforcement learning (MARL) setting with multiple discrete actions and a global constraint on the state space. This MARL setting is suitable for many industrial tasks such as inventory management where each agent chooses a replenishing quantity level for the corresponding stock-keeping-unit (SKU) such that the total profit is maximized while the total inventory does not exceed a certain limit. Based on WIMS, we propose a deep MARL algorithm called WIMSN for inventory management. Empirically, WIMSN is computationally efficient (without the need to re-train the model when the constraint limit or the combinition of SKUs changes), scalable (to thousands of agents), and high-performing (surpassing previous OR and MARL baselines especially on hard tasks with tight constraints).

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

# A INVENTORY MANAGEMENT

We consider a simplified setting with one store and multiple SKUs to better focus on the constrained MARL setting considered in this paper. We further assume that there is an upstream warehouse that can fulfill all the requirements from the store. Our objective is to train a high-quality replenishing policy for each SKU in the store, particularly when there are a large number of SKUs and when the constraint on the inventory capacity is tight. Moreover, due to the flexibility of RL algorithms, our method can also be applied to more complex settings, e.g., with multi-echelon structure or fluctuate supply.

Similar to previous work, we follow the multi-agent RL (MARL) framework with decentralized agents, each of which manages the inventory of one SKU in the store. We assume that the store has $N$ SKUs in sell, all of which share a common inventory capacity that can store up to $L_t$ units at the same time.

Replenishing decisions of each SKU are made on discretized time steps (which are days in this paper). For the $t$-th time step and the $i$-th SKU, we denote the in-stock quantity by $I_t^i \in \mathbb{Z}$. This quantity is constrained by the limit of the inventory capacity (i.e., the shared resources). Accordingly, the following constraint holds:

$$\forall t \geq 0, \sum_{i=1}^{n} I_t^i \leq L_t. \tag{7}$$

The dynamics of the system is as follows: For the $t$-th time step and the $i$-th SKU, the corresponding agent can place a replenishment order to request $O_t^i \in \mathbb{Z}$ units of products from its upstream warehouse. It takes several time steps (referred to as the leading time $LT_t^i$) before these products are delivered to the store. We denote the total units in transit at the $t$-th time step by $T_t^i \in \mathbb{Z}$ and the total units arrived at the $t$-th time step by $A_t^i \in \mathbb{Z}$. Meanwhile, the customer demand is $D_t^i$ and leads to an actual sale of $S_t^i \in \mathbb{Z}$ units. Due to the possibility of out-of-stock, $S_t^i$ may be smaller than $D_t^i$. We consider the setting where the leading time $LT_t^i$ and the demand $D_t^i$ are stochastic.

For clearness, we summarize the dynamics these variables as follows:

$$S_t^i = \min\left(D_t^i, I_t^i\right) \qquad\qquad A_t^i = \sum_{\tau : \tau + LT_\tau^i = t} O_\tau^i$$

$$T_{t+1}^i = T_t^i - A_{t+1}^i + O_{t+1}^i \qquad\qquad I_{t+1}^i = I_t^i - S_t^i + A_{t+1}^i$$

The immediate profit of the $i$-th SKU is calculated according to the following equation:

$$R_t^i = p_i S_t^i - q_i O_t^i - o_i \mathbb{I}\left[O_t^i > 0\right] - b_i (D_t^i - S_t^i) \tag{8}$$

where $p_i$, $q_i$, $o_i$, $b_i$ are the unit sale price, the unit procurement price, the order cost, and the unit backlog cost for the $i$-th SKU respectively, and $\mathbb{I}[\cdot]$ is an indicator function which equals to one when the condition is true and zero otherwise. The order cost reflects the fixed transportation cost or the order processing cost, and yields whenever the order quantity is non-zero.

The objective of this task is to find a policy $\pi$ that generates replenishment orders $O_t^i$ for each SKU to solve the following constrained optimization problem:

$$\max_{\pi} \mathbb{E}\left[\sum_{t=0}^{\infty}\sum_{i=1}^{N} \gamma^t R_t^i \Big| \pi\right], \qquad s.t., \sum_{i=1}^{N} I_t^i \leq L_t, \forall t. \tag{9}$$

When the total in-stock quantity $\sum_{i=1}^{N} I_t^i$ exceeds the inventory capacity $L_t$, we reduce equal amount of in-stock quantities of different SKUs such that the total in-stock quantity equals the capacity limit. In the train procedure of baseline algorithms, we calculate the reward penalty for the dropped quantities before dropping them, and use the with-penalty reward to train the model. In the test procedure, the dropped quantities do not contribute any reward (neither income nor penalty), just as they have never been replenished. In addition, we will also record the quantity of constraint violation $\text{vio}_t := \sum_{i=1}^{N} I_t^i - L_t$ and report the maximum constraint violation $\text{VIO} = \max_t \text{vio}_t$.

**Remark.** We formulate the inventory management problem as an optimization problem with hard constraints. However, from a practical perspective, it is unlikely to strictly prevent constraint violation

| Notation | Explanation |
|---|---|
| $N$ | The number of SKUs |
| $I_t^i$ | Inventory of the $i$-th SKU at the $t$-th time step |
| $O_t^i$ | Order quantity of the $i$-th SKU at the $t$-th time step |
| $D_t^i$ | Demand of the $i$-th SKU at the $t$-th time step |
| $LT_t^i$ | Lead time of the order placed at the $t$-th time step for the $i$-th SKU |
| $S_t^i$ | Sale quantity of the $i$-th SKU at the $t$-th time step |
| $T_t^i$ | Quantity in transit of the $i$-th SKU at the $t$-th time step |
| $R_t^i$ | Profit generated on the $i$-th SKU at the $t$-th time step |
| $p_i$ | Unit sales price of the $i$-th SKU |
| $q_i$ | Unit procurement cost of the $i$-th SKU |
| $o_i$ | Unit order cost of the $i$-th SKU |
| $b_i$ | Unit backlog cost of the $i$-th SKU |
| $L$ | Storage capacity of store |

Table 2: Notation for the inventory management problem considered in this paper.

(exceeding the capacity limit) w/o reducing the profitability. Specifically, there are many practical workarounds when the inventory overflows, such as clearance sale, renting additional storage space, or returning products, and these workarounds are preferable than using an overly conservative and low-profitability strategy. Therefore, our objective under this mathematical formulation is to obtain a policy such that this constraint strictly holds in most of the cases while achieving as high profit as possible.

For convenience, we summarize the notations used in this paper in Table 2.

# B  PROOF

## B.1  PROOF OF LEMMA 4.2

Lemma 4.2 (action monotonicity to indexability) can be proved by contradiction as follows: We assume that the action monotonicity condition is satisfied but there exists an $\alpha$ such that $\mathbb{S}_\alpha(\lambda)$ does not decrease monotonically (i.e., the agent is not indexable). This indicates that there exists a state $s \in \mathcal{S}$ and $\lambda_1 < \lambda_2$ such that $s \notin \mathbb{S}_\alpha(\lambda_1)$ but $s \in \mathbb{S}_\alpha(\lambda_2)$. In other words, when $\lambda = \lambda_1$, the optimal policy selects an action $a \leq \alpha$ on state $s$; when $\lambda = \lambda_2$, the optimal policy selects an action $a > \alpha$ on the same state. Therefore, we have $\pi_{\lambda_1}^*(s) \leq \alpha < \pi_{\lambda_1}^*(s)$ which contradicts the assumption that $\pi_{\lambda_1}^*(s) \geq \pi_{\lambda_1}^*(s)$.

## B.2  FEASIBILITY OF ACTION MONOTONICITY CONDITION IN LEMMA 4.2

In this section, we provide an intuition why this assumption is natural for inventory management. By defining $I(\pi) := \mathbb{E}[\sum_{t=0}^\infty \gamma^t l(s_t)|\pi]$ where the expectation is over the initial state distribution and the randomness of the environment/policy, we can decompose $Q_\lambda(\pi) = J(\pi) - \lambda I(\pi)$.

We can prove $I(\pi_{\lambda_1}^*) \leq I(\pi_{\lambda_2}^*)$ if $\lambda_1 > \lambda_2$. This indicates that we would better to maintain a higher inventory level when $\lambda$ decreases, which in turn requires us to take larger actions to order more quantities for replenishment.

**Lemma B.1.** $I(\pi_{\lambda_1}^*) \leq I(\pi_{\lambda_2}^*)$ if $\lambda_1 > \lambda_2$

*Proof.* If $I(\pi_{\lambda_1}^*) > I(\pi_{\lambda_2}^*)$, then

$$(\lambda_1 - \lambda_2)[I(\pi_{\lambda_1}^*) - I(\pi_{\lambda_2}^*)] = \lambda_1 I(\pi_{\lambda_1}^*) - \lambda_2 I(\pi_{\lambda_1}^*) - \lambda_1 I(\pi_{\lambda_2}^*) + \lambda_2 I(\pi_{\lambda_2}^*) > 0,$$

i.e.,

$$\lambda_1 I(\pi_{\lambda_1}^*) - \lambda_2 I(\pi_{\lambda_1}^*) - \lambda_1 I(\pi_{\lambda_2}^*) > -\lambda_2 I(\pi_{\lambda_2}^*)$$

Thus,

$$
\begin{aligned}
Q_{\lambda_2}(\pi_{\lambda_1}^*) &= J(\pi_{\lambda_1}^*) - \lambda_2 I(\pi_{\lambda_1}^*) \\
&= J(\pi_{\lambda_1}^*) - \lambda_1 I(\pi_{\lambda_1}^*) + (\lambda_1 - \lambda_2) I(\pi_{\lambda_1}^*) \\
&\geq J(\pi_{\lambda_2}^*) - \lambda_1 I(\pi_{\lambda_2}^*) + (\lambda_1 - \lambda_2) I(\pi_{\lambda_1}^*) \\
&= J(\pi_{\lambda_2}^*) - \lambda_1 I(\pi_{\lambda_2}^*) + \lambda_1 I(\pi_{\lambda_1}^*) - \lambda_2 I(\pi_{\lambda_1}^*) \\
&> J(\pi_{\lambda_2}^*) - \lambda_2 I(\pi_{\lambda_2}^*) \\
&= Q_{\lambda_2}(\pi_{\lambda_2}^*),
\end{aligned}
$$

which contradicts the fact that $\pi_{\lambda_2}^*$ is the optimal policy for $\lambda_2$. $\qquad\square$

### B.3 PROOF OF LEMMA 4.5

The following proof goes for any $s \in \mathcal{S}$ and $\alpha \in [A - 1]$ and an arbitrary set of optimal policies $\pi_\lambda^*$ for $\lambda \in \mathbb{R}$ that define $\mathbb{S}_\alpha$ and $f(s)$.

1) If $\lambda < [f(s)]_\alpha := \sup_\lambda \{\lambda : s \in \mathbb{S}_\alpha(\lambda)\}$, we have $\lambda = [f(s)]_\alpha - \epsilon$ for some $\epsilon > 0$. By Definition 4.1, when $\epsilon > 0$, $s \in \mathbb{S}_\alpha(\lambda)$ and therefore the optimal policy $\pi_\lambda^*$ satisfies $\pi_\lambda^*(s) > \alpha$, i.e., $Q_\lambda^*(s, > \alpha) \geq Q_\lambda^*(s, \leq \alpha)$.

2) Similarly, if $\lambda > [f(s)]_\alpha$, we have $s \notin \mathbb{S}_\alpha(\lambda)$ and therefore the optimal policy $\pi_\lambda^*$ satisfies $\pi_\lambda^*(s) \leq \alpha$, i.e., $Q_\lambda^*(s, > \alpha) \leq Q_\lambda^*(s, \leq \alpha)$.

3) If $Q_\lambda^*(s, > \alpha)$ and $Q_\lambda^*(s, \leq \alpha)$ is continuous w.r.t. $\lambda$, it is not hard to see $Q_\lambda^*(s, > \alpha) = Q_\lambda^*(s, \leq \alpha)$ when $\lambda = [f(s)]_\alpha$ by combining the previous two arguments.

Next, we prove the continuity of $Q_\lambda^*(s, a)$ which easily leads to the continuity of $Q_\lambda^*(s, > \alpha)$ and $Q_\lambda^*(s, \leq \alpha)$ since the maximum over several continuous functions is continuous. Suppose $\pi_1$ and $\pi_2$ are the optimal policies when $\lambda = \lambda_1$ and $\lambda = \lambda_2$ respectively. Due to optimality, we have $Q_{\lambda_2}^{\pi_2}(s, a) \geq Q_{\lambda_2}^{\pi_1}(s, a)$ and $Q_{\lambda_1}^{\pi_1}(s, a) \geq Q_{\lambda_1}^{\pi_2}(s, a)$ for any $(s, a)$. By the definition of $Q_\lambda^\pi(s, a)$ in (1), $Q_\lambda^\pi(s, a)$ is continuous w.r.t. $\lambda$ for any $(s, a)$, i.e., for given $\epsilon > 0$, there exists $\delta > 0$ such that $|\lambda_1 - \lambda_2| \leq \delta \Rightarrow |Q_{\lambda_2}^{\pi_2}(s, a) - Q_{\lambda_1}^{\pi_2}(s, a)| \leq \epsilon$ and $|Q_{\lambda_2}^{\pi_1}(s, a) - Q_{\lambda_1}^{\pi_1}(s, a)| \leq \epsilon$. Therefore, we have $Q_{\lambda_2}^{\pi_2}(s, a) - Q_{\lambda_1}^{\pi_1}(s, a) \leq Q_{\lambda_2}^{\pi_2}(s, a) - Q_{\lambda_1}^{\pi_2}(s, a) \leq \epsilon$ and $Q_{\lambda_2}^{\pi_2}(s, a) - Q_{\lambda_1}^{\pi_1}(s, a) \geq Q_{\lambda_2}^{\pi_1}(s, a) - Q_{\lambda_1}^{\pi_1}(s, a) \geq -\epsilon$, which imply $|Q_{\lambda_2}^{\pi_2}(s, a) - Q_{\lambda_1}^{\pi_1}(s, a)| \leq \epsilon$ for any $(s, a)$ and completes the proof.

### B.4 PROOF OF LEMMA 4.6

When $L_t \to \infty$, we want to show that the WIMS policy with the global cost $\lambda_g = 0$ is optimal. This policy can be written as

$$
\begin{aligned}
a_t^i &= \sum_{\alpha < A} \mathbb{I}[[f(s_t^i)]_\alpha \geq 0] + 1 \\
&= \sum_{\alpha < A} \mathbb{I}[Q_0^*(s, > \alpha) \geq Q_0^*(s, \leq \alpha)] + 1 \\
&= \arg\max_\alpha Q_0^*(s, \alpha)
\end{aligned}
\tag{10}
$$

where the first equation follows the definition of the WIMS policy (Eq. 3), the second equation follows Lemma 4.5, the third equation can be derived following the definition in Eq. 2. Notice that when the capacity limit is infinite, the optimal policy should always take the action $\arg\max_\alpha Q_{\lambda=0}^*(s, \alpha)$.

## C IMPLEMENTATION

In addition to the WIMSN-c (that learns the WIMS policy with neural network on continuous $\lambda$) presented in Algorithm 1. We also design another version called WIMSN-d (that learns the WIMS policy with neural network on $K$ discretized $\lambda_1, \cdots, \lambda_K$) and present it in Algorithm 2. With prescribed grids of $\lambda$s, WIMSN-d decouples the inter-dependency for learning the Q network and learning the Whittle index network, and therefore is more stable. However, the performance of WIMSN-d may be limited by the discretization of $\lambda$ and this explains why WIMSN-c outperforms

---

**Algorithm 2** WIMSN-d: Learn the WIMS policy with neural network on **d**iscretized $\lambda$s

1: Given $\Lambda := (\lambda_1, \cdots, \lambda_K)$
2: **Initialize:**
3:    The Q network $Q_\theta : \mathcal{S} \to \mathbb{R}^{A \times K}$ with the $(a, k)$-th element representing $Q_\theta(s, a, \lambda_k)$
4:    The Whittle index network $f_\phi : \mathcal{S} \to \mathbb{R}^{A-1}$
5: ▷ *Phase 1: Train agents independently without any constraint*
6: Initialize the replay buffer $\mathcal{D}$ with warming-up samples collected by the random policy
7: **for** each iteration **do**
8:    Select $\lambda \leftarrow \lambda_k$ for some $k$ sampled from $Unif([K])$
9:    Roll out the $\epsilon$-greedy version of the policy $\mu_\lambda(\cdot) = \arg\max_a Q_\theta(\cdot, a, \lambda)$
10:    Store new transitions to the replay buffer $\mathcal{D}$
11:    Sample a batch $\mathcal{B} := \{(s, a, r, s')\}$ from $\mathcal{D}$
12:    Update the Q-network following (4)
13:    Generate Whittle index labels for the batch $\{(s, \hat{\mathbf{f}})\}$
14:    Update the Whittle index network by SGD on $\min_\phi \mathbb{E}_{\mathcal{B}}[\|f_\phi(s) - \hat{\mathbf{f}}\|^2]$
15: **end for**
16: ▷ *Step 2: Train joint agents*
17: The same as Algorithm 1

---

WIMSN-d on tasks with over 1000 SKUs (cf. Table 1). We provide the source code in `https://github.com/zhangchuheng123/WIMSN`.

For OR-based methods and two previous MARL-based methods (IPPO and IDQN), the optimization objective in the train procedure is the cumulative profit plus a reward penalty equal to 50% of the procurement cost of dropped products, which defines a consistent notion of optimality in these baselines.

## C.1 IMPLEMENTATION OF OR-BASED METHODS

We select two popular OR algorithms, the base-stock policy and the $(s, S)$-policy, as our operational-research-based benchmarks.

For each SKU, the $(s, S)$-policy has two parameters: $s_{\text{param}}$ is the re-order point and $S_{\text{param}}$ is the order up-to level. On each day, the policy places an order to replenish the inventory level up to the order up-to level $S_{\text{param}}$ if the quantity of the order exceeds the re-order point $s_{\text{param}}$. Specifically, the policy places an order with the quantity

$$O = \begin{cases} S_{\text{param}} - I - T & S_{\text{param}} - I - T \geq s_{\text{param}} \\ 0 & \text{otherwise} \end{cases}, \tag{11}$$

where $O$ is the quantity of the replenishment order, $I$ is the inventory level, $T$ is the quantity in transit.

The base stock policy is a special case of the $(s, S)$-policy with $s_{\text{param}} = 0$.

For a task with $N$ SKUs, the $(s, S)$-policy and the base stock policy need to specify $2N$ and $N$ parameters respectively. Consequently, it is not trivial to obtain the best set of parameters on a given task, and we use genetic algorithm (GA) to search for these best parameters. First, we search for the best set of shared parameters (i.e., using the same parameters for all the SKUs) and use it as the initial solutions in GA. Second, we search using GA with the population size as 100 and the minimum number of iterations as 1000. Third, we terminate the search only if the minimum number of iterations is reached as well as the solution is a local optimum. To verify whether the solution is a local optimum, we evaluate the performance of all the element-wise perturbations to this solution and treat the solution as a local optimum only if the solution performs the best among these perturbed candidates.

The evaluation metric (in the train procedure) of these OR-based methods is the same as the baseline MARL algorithms for consistency, i.e., the cumulative profit plus a penalty equal to 50% of the procurement cost of overflowed products. Indeed, we can trade off the constraint violation with the profit by adjusting for the coefficient. However, we find that using different coefficients does not influence the dominant advantage of our algorithm.

| Hyperparameters | Values |
|---|---|
| The number of training iterations | $5 \times 10^6$ |
| The number of episodes collected in each iteration | 8 |
| The number of time steps in each episode | 100 |
| The capacity (in #episodes) of the replay buffer | 5000 |
| $\epsilon$ annealing scheme for the $\epsilon$-greedy exploration | Linear annealing |
| $\epsilon$ annealing span for the $\epsilon$-greedy exploration | $4 \times 10^6$ iterations |
| $\epsilon$ values for the $\epsilon$-greedy exploration | From 1.0 to 0.05 |
| The number of layers in the Q network | 4 |
| The number of neurons in each hidden layer of the Q network | 128 |
| The number of layers in the Whittle index network | 4 |
| The number of neurons in each hidden layer of the Whittle index network | 128 |
| Discount rate $\gamma$ | 0.985 |
| Optimizer | PMSProp |
| Learning rate for the Q network | $5 \times 10^{-4}$ |
| Learning rate for the Whittle index network | $1 \times 10^{-4}$ |
| Gradient clip | 10 |

Table 3: The hyperparameters of WIMSN

## C.2 IMPLEMENTATION OF BASELINE MARL ALGORITHMS

For these MARL algorithms that do not follow the constrained MDP formulation, we penalize the violation of the constraint into the immediate profit to help the algorithms account for the constraint. Specifically, if $\sum_{i=1}^{N} I_t^i > L$ for some time step $t$, we penalize each agent (indexed by $j$) with an additional cost which equals to $0.5 \times \frac{1}{N}(\sum_{i=1}^{N} I_t^i - L) \times q_j$ where $q_j$ is the unit procurement price of the $j$-th SKU. We train the agents using individual rewards (IR) which provides penalty-regularized reward to each agent, and we let all the agents amortize the overflowing penalty [2].

As mentioned before, in the train procedure, these MARL baselines use the reward function as the cumulative profit plus a penalty equal to 50% of the procurement cost of overflowed products. This penalty coefficient trades off the constraint violation with the profit. We observe that these MARL baselines result in signification constraint violation, which calls for larger penalty coefficient. However, we find that further increasing the coefficient will lead to instability, and this is the reason why we set the coefficient to 50%.

For the MARL baselines and our WIMSN implementation, we additionally use the following techniques to improve the stability and the generalization of the model:

- Observation normalization: Since different SKUs may vary greatly in prices, quantities, and demands, we normalize different features in the observation individually. Specifically, we divide the features with price/quantity/demand as the unit with the selling price/the storage capacity/the average demand over the past 21 days respectively. For example, the feature "in-stock quantity" is equal to 500 with the "quantity" unit and the storage capacity is 5000. Then, we normalize the feature to $500/5000 = 0.1$.

- Action normalization: Since different SKUs may vary greatly in the range of the demands, we use the quantity of the replenishment orders as the multiply of the action and the average demand of the corresponding SKU over the past 21 days. This greatly improve the generalization of the model over different SKUs.

- State/reward normalization: We calculate the standard score (or z-score) (Huck et al., 1986) of the state and the reward using the rolling average and standard deviation during the training.

- Loss normalization: For WIMSN-d, since the Q values for different $\lambda$s have different value ranges, we normalize the loss separately for different dimensions of $\lambda$.

---

[2]We have also tried to use team reward (TR) which is free of the penalty assignment problem. However, the experiment results suggested that using IR is better than or equal to using TR.

| Testing | MARL-Based Methods | | | | |
| Performance | Baselines | | | Ours | |
| | IPPO | IDQN | Primal-Dual | WIMSN-d | WIMSN-c |
| N100-C2.5k | 391 | 435 | 420 | | |
| N100-C5k | 400 | 432 | 423 | | |
| N100-C10k | 390 | 432 | 428 | | |
| N100-C20k | 397 | 430 | 425 | | |
| N1k-C25k | 1222 | 1959 | 1207 | 2652 in total | 2575 in total |
| N1k-C50k | 1228 | 2061 | 1247 | (One model | (One model |
| N1k-C100k | 1227 | 2061 | 1183 | for all tasks) | for all tasks) |
| N1k-C200k | 1190 | 2021 | 1253 | | |
| N2k-C58k | 1567 | 2728 | 1602 | | |
| N2k-C115k | 1618 | 2858 | 1689 | | |
| N2k-C231k | 1593 | 2847 | 1549 | | |
| N2k-C461k | 1548 | 2703 | 1612 | | |

Table 4: The computational cost of different algorithms in different inventory management tasks with different numbers of SKUs (indicated by "N") and capacity limits (indicated by "C"). See detailed description of the algorithms in comparison in Section 6. We present the average wall time (in minutes) of all the corresponding runs in Table 1. Note that in WIMSN-c or WIMSN-d, we use the same model for all different tasks with different numbers of SKUs and capacity limits, this greatly reduces the computational costs of WIMSN-c and WIMSN-d.

## C.3 IMPLEMENTATION OF WIMSN-C

For WIMSN-c and WIMSN-d, we only train one model (i.e., the Whittle index network) on all 2307 SKUs and use it for evaluation on all tasks with different combinations of SKUs and capacity limits. WIMSN-c is implemented based on the code of the IDQN implementation, and we list the hyperparameters in Table 3.

Thanks to Proposition 4.4, the outcome of this rule is equivalent to the rule (3) we used in our algorithm. However, we find that (3) is more robust since the output of the Whittle index network may sometimes violate Proposition 4.4 slightly and it results in better empirical performance.

When initializing the Whittle index network, we add a constant offset $\lambda_{const} = 4$ to the output. The offset is chosen as a reasonable prior for the cost $\lambda$ which should correspond to the maximum affordable unit storage in a typical supply chain. This trick makes the networks are trained initially over a reasonable range of $\lambda$s.

## C.4 IMPLEMENTATION OF WIMSN-D

We use $K = 51$ and $(\lambda_1, \cdots, \lambda_K) = $ `np.linspace(0, 10, 51)`.

Given a state $s$, the Q network outputs a matrix $Q \in \mathbb{R}^{A \times K}$. The Whittle index label for this sample $\hat{\mathbf{f}}$ can be calculated as follows:

```
Q_max_forward = np.maximum.accumulate(Q, axis=0)
Q_max_backward = np.maximum.accumulate(Q[::-1], axis=0)[::-1]
Q_diff = np.abs(Q_max_forward - Q_max_backward)
f̂ = np.argmin(Q_diff, axis=1)[:-1]
```

# D EXTENDED EXPERIMENT RESULTS

We show the computational costs (in terms of the wall time in minutes) of different algorithms in Table 4. We run our experiments on a machine with $24\times$ Intel Xeon CPU E5-2690 v4 CPU, $1\times$ NVIDIA V100 16G GPU, and 512G memory. We also show the standard deviation of the daily profit per SKU for different MARL-based algorithms in Table 5.

This dataset used in this paper is provided by one of our retail partners. The experiments are based on the history of 2307 SKUs from 2018-08-01 to 2019-12-13. The training / validation / testing sets are set to be the subset 2018-08-01 to 2019-05-27 (300 days) / 2019-05-28 to 2019-09-04 (100 days)

| Testing Performance | MARL-Based Methods | | | | |
|---|---|---|---|---|---|
| | Baselines | | | Ours | |
| | IPPO | IDQN | Primal-Dual | WIMSN-d | WIMSN-c |
| N100-C2.5k | 0.07 | 0.06 | 0.17 | 0.03 | 0.03 |
| N100-C5k | 0.39 | 0.41 | 0.85 | 0.15 | 0.20 |
| N100-C10k | 0.07 | 0.08 | 0.07 | 0.01 | 0.07 |
| N100-C20k | 0.01 | 0.07 | 0.08 | 0.00 | 0.07 |
| N1k-C25k | 0.46 | 0.43 | 0.36 | 0.14 | 0.15 |
| N1k-C50k | 0.67 | 0.78 | 0.93 | 0.23 | 0.40 |
| N1k-C100k | 0.44 | 0.02 | 0.38 | 0.10 | 0.36 |
| N1k-C200k | 0.23 | 0.34 | 0.20 | 0.14 | 0.10 |
| N2k-C58k | 1.67 | 0.98 | 0.50 | 0.08 | 0.01 |
| N2k-C115k | 1.28 | 1.42 | 0.80 | 0.39 | 0.46 |
| N2k-C231k | 0.47 | 0.33 | 0.15 | 0.45 | 0.27 |
| N2k-C461k | 0.13 | 0.30 | 0.01 | 0.18 | 0.19 |

Table 5: The standard deviation across different seeds on the daily profit per SKU for different algorithms in different inventory management tasks with different number of SKUs (indicated by "N") and capacity limit (indicated by "C").

/ 2019-09-05 to 2019-12-13 (100 days) respectively. For the experiments on 100/1000 SKUs, we sample a subset of SKUs uniformly randomly from the 2307 SKUs.

