# OpenReview forum: "Whittle Index with Multiple Actions and State Constraint for Inventory Management"
_ICLR.cc/2024/Conference — ICLR 2024 poster_

### Official Review · Reviewer_qxXR · 2023-10-30

**Soundness:** 3 good
**Presentation:** 3 good
**Contribution:** 2 fair
**Rating:** 5
**Confidence:** 3

**Summary:**

This paper is about computational studies of a Whittle index-based policy on a variant of the restless bandit problem. The restless bandit setting allows multiple actions for each bandit, and there is a constraint that controls the joint state of the bandits. In particular, the paper focuses on the inventory management of multiple stock-keeping units. Here, each stocking-keeping unit corresponds to a bandit, and the list of stock levels that a unit can maintain is the set of actions. The distinction with the existing works on restless bandits is that the problem is not about selecting one bandit but about choosing an action for each bandit. That said, it makes sense to define and compare the indices of multiple actions, instead of comparing some indices of distinct bandits. Moreover, the inventory level constraint imposes restrictions on taking which actions to different units. Therefore, the problem can be viewed as an instance of multi-agent reinforcement learning (MARL).

**Strengths:**

* The paper provides a novel definition of a Whittle index-type policy for a multi-agent inventory management problem. Indexability for the setting is defined, and some sufficient conditions for the notion of indexability are provided. The framework seems novel.
* In general, implementing a Whittle index policy can be inefficient as computing Whittle indices is difficult. However, numerical results show that the proposed Whittle index-based policy can be efficiently implemented and at the same time, one can impose satisfying the joint inventory level constraint at least computationally.

**Weaknesses:**

* No theoretical guarantee on the proposed method is provided. It seems that the framework sits between restless bandits and multi-agent reinforcement learning (MARL). That said, the reader would wonder if any theoretical results on either restless bandits or MARL extend to the particular problem setting of this paper.
* It is not clear how the joint inventory level constraint is satisfied by the WIMS policy. The WIMS policy controls individual stock-keeping units separately while the only joint control is on updating the dual variable. That said, in principle, one may use any algorithm for dealing with individual units. It is difficult to convince that the WIMS policy is particularly effective for the multi-agent setting studied in this paper.

**Questions:**

* Is it possible to prove that without the joint inventory level constraint, the Whittle index-based policy of this paper achieves optimality?
* Is it possible to discover and explain any connection between the setting of this paper and the MARL settings, e.g., fully competitive and cooperative settings?
* Is it possible to compare the proposed framework of this paper and the existing single-agent inventory control methods combined with dual penalization as done in this paper?

---

> ### Author Response · Authors · 2023-11-20
> **Author Response to Reviwer qxXR**
>
> Dear Reviewer,
>
> We greatly appreciate your insightful comments and the time you took to review our manuscript. Your feedback is essential to improving our work, and here we address your concerns:
>
> **No theoretical guarantee on the proposed method is provided. It seems that the framework sits between restless bandits and multi-agent reinforcement learning (MARL). That said, the reader would wonder if any theoretical results on either restless bandits or MARL extend to the particular problem setting of this paper.**
>
> Whittle index is an effective tool for restless bandits but lacks theoretical guarantee (only asymptotic results exist). Following this line of work, we leverage the Whittle index as a tool to tackle the large-scale multi-agent system and find it is hard to obtain rigorous theoretical results. Thanks to Q1 raised by the reviewer, we obtain an asymptotic optimality result (see the response to Q1 below). We also want to emphasize  that our paper aims to look for a good learning policy in practice. It would be really appreciated if the reviewer could evaluate our paper more based on the empirical performance.
>
> **It is not clear how the joint inventory level constraint is satisfied by the WIMS policy. The WIMS policy controls individual stock-keeping units separately while the only joint control is on updating the dual variable. That said, in principle, one may use any algorithm for dealing with individual units. It is difficult to convince that the WIMS policy is particularly effective for the multi-agent setting studied in this paper.**
>
> The effectiveness of the WIMS policy is based on the following principle: When each agent generates an individual unit (scalar) to indicate the maximum affordable costs the agent is willing to pay for the shared resources (e.g., the inventory capacity), these agents can be well coordinated by a global dual variable indicating the global pricing for the shared resources. Therefore, the algorithms that generate other forms of the individual units may not result in an effective strategy.
>
> **Q1: Is it possible to prove that without the joint inventory level constraint, the Whittle index-based policy of this paper achieves optimality?**
>
> Yes, we can prove this asymptotic ($L_t \to \infty$) optimality. The proof is as follows: When $L_t \to \infty$, we want to show that the WIMS policy with the global cost $\lambda_g = 0$ is optimal. This policy can be written as
>
> $$ a_t^i = \sum_{\alpha < A} \mathbb{I} [  [f(s_t^i)]_\alpha \ge 0 ] + 1 $$
>
> $$ = \sum_{\alpha < A} \mathbb{I} [  Q_0^*(s, >\alpha) \ge Q_0^*(s, \le \alpha) ] + 1 $$
>
> $$ = arg \max_\alpha Q_0^*(s, \alpha) $$
>
> where the first equation follows the definition of the WIMS policy (Eq. 3), the second equation follows Lemma 4.5, the third equation can be derived following the definition in Eq. 2. Notice that when the capacity limit is infinite, the optimal policy should always take the action $arg \max_\alpha Q_{\lambda = 0}^*(s, \alpha)$.
>
> **Q2: Is it possible to discover and explain any connection between the setting of this paper and the MARL settings, e.g., fully competitive and cooperative settings?**
>
> Our paper considers a fully cooperative MARL setting (since all agents share the same objective) with a global constraint on the state space.
>
> **Q3: Is it possible to compare the proposed framework of this paper and the existing single-agent inventory control methods combined with dual penalization as done in this paper?**
>
> We would be grateful if the reviewer could provide additional clarification on this question. For a single agent system, using the dual penalization may not be necessary since we can adjust the policy based on the inventory excess directly. We guess the reviewer refers to the "Primal-Dual" algorithm [Chen et al 2021] mentioned in our paper. The difference between WIMSN and Primal-Dual is that Primal-Dual trains a policy network directly that is not adaptive for other constraint levels. In contrast, by leveraging the concept of the Whittle index, WIMSN trains a Whittle index network that is adaptive to different constraint levels.

---

> > ### Comment · Reviewer_qxXR · 2023-11-22
> >
> > Thank you for your responses to the comments. However, I found that the author's responses are not satisfactory, so I would stick to the current score.
> >
> > For the first weakness part, as the authors mentioned, there exist some asymptotic optimality results on restless bandits for the case of no constraint. That said, one would be interested in whether WIMS is asymptotically optimal for the constrained case. As noted in the first review report, there is no theoretical guarantee.
> >
> > For the second weakness part, again, each agent can use any inventory control policy while the agents can still be coordinated by a global dual variable. That said I disagree with the comment that other algorithms would not result in an effective strategy.
> >
> > For the first question, I appreciate your argument for the optimality without the joint inventory level constraint. However, the presence of the joint constraint is what makes the WIMS policy suboptimal perhaps unless there is a rigorous theoretical guarantee for the case of finite $L_t$.
> >
> > For the second question, I do not agree with your statement that the problem is simply fully cooperative given that the agents should somehow compete due to the joint inventory level restriction.
> >
> > For the third question, this question is related to the second weakness point.

---

> > > ### Author Response · Authors · 2023-11-23
> > > **Thanks for the Further Feedback**
> > >
> > > Thanks for your further feedback.
> > >
> > > In our response to the second weakness, what we would like to say is that our formulation is a natural algorithm to result in effective strategies. In other words, defining the global dual variable to be the unit storage cost and subsequently deriving the index-conditioned inventory control policy is a natural choice. Other inventory control policies, e.g., traditional inventory control strategies such as the base-stock policy, cannot be easily coordinated by a global dual variable.
> > >
> > > For the second question, in the fully coorperative setting, all the agents have the same target. In other words, there is no concept about the profit of individual agents, and therefore there is no individual objective for different agents to compete for. For example, an SKU with a low profit level may decrease its in-stock to let the SKUs with a high profit level order more units to get more total profit.

---

> > > ### Author Response · Authors · 2023-11-23
> > > **Further Explanation on the Second Weakness**
> > >
> > > Let's use some examples to illustrate our points about W2.
> > >
> > > 1) Due to the format of the optimization problem, the only meaningful global dual variable is the unit storage cost (as in our paper). Of course one can choose other kinds of global dual variable to try to limit the total in-stock level. However, in this case, their optimization target could become far from the origin one. Let's consider using the activation cost as the global dual variable (as traditional Whittle Index). In this case, one can figure out that the index of a local state for some SKU is almost independent with its in-stock level, but depend highly on its profit level. In fact, if a SKU has a very high profit level, then its index is always higher than the other SKUs with low profit level, even if its in-stock level is extremely large and the other SKUs' in-stock level is very small. In this case, if we use a global dual variable to control the total in-stock level, then we will always order the SKU with the highest profit level as many as we can, and do not order the other SKUs even if we are running out them in the store. This is obviously an non-effective policy, since when the in-stock level of the SKU with the highest profit level is enough to avoid backlogging, ordering some other SKUs would be better.
> > >
> > > ii) Once the global dual variable is the unit storage cost, the individual reward function is $Q^\lambda(s,a) = R - \lambda * s$. In this case, for a local state $s$, and a global dual variable $\lambda_g$, the only meaningful action is to choose $a = arg\max Q^{\lambda_g}(s,a)$, and this is our algorithm. If we look for other kinds of algorithms, e.g., changing $Q^\lambda(s,a) = R - \lambda * s$ to $Q'^\lambda(s,a) = R' - \lambda * s$ with a totally different reward function R', then this one may be able to control the total in-storage level, but this optimization target is $R'$, which is far from $R$. This means it is also not an effective algorithm.

---

> ### Author Response · Authors · 2023-11-22
> **Thank You for the Feedback**
>
> Dear Reviewer qxXR,
>
> As the author reviewer discussion period is ending soon, we sincerely thank you for the invaluable feedback. Should our response effectively address your concerns, we kindly hope that you could consider raising the score rating for our work. We will also be happy to address any additional queries or points.
>
> Best regards,
>
> Authors

---

### Official Review · Reviewer_xHVA · 2023-10-31

**Soundness:** 2 fair
**Presentation:** 2 fair
**Contribution:** 2 fair
**Rating:** 3
**Confidence:** 4

**Summary:**

This paper considered a multi-agent reinforcement learning setting with multiple actions and one coupling state constraint which has broad applicability across various fields, including inventory management.  The authors proposed WIMS by leveraging deep multi-agent reinforcement learning. Specifically, WIMS built on top of the state-of-the-art Whittle index policy, and generalize it to multi-action and dynamic state constraint settings. Experimental results were provided to validate the performance of WIMS.

**Strengths:**

This paper considered a multi-agent reinforcement learning setting with multiple actions and one coupling state constraint which has broad applicability across various fields, including inventory management.  The authors proposed WIMS by leveraging deep multi-agent reinforcement learning.

**Weaknesses:**

1. In this paper, the authors assume that the multiple SKUs are independent and their total inventory level cannot exceed some capacity constraint. Unfortunately, this assumption is problematic and lack of justification. For example, rewards are not only independent in large-scale systems (a large number of SKU), and the substitution effect can occur.

2. The proposed algorithm WIMS is mostly heuristic based and lack performance guarantees. First, this paper introduces a state constraint that must be satisfied in each time step $t$ in Section 4.1. However, WIMS is NOT guarantee to satisfy this constraint at all, and hence there is a gap between the formulated problem and the proposed solution. Second, it is not quite clear if the proposed WIMS is asymptotically optimal or not, which is a key performance in restless bandits literature. For example, the state-of-the-art Whittle index policy is provably asymptotically optimal. However, Whittle index policy is designed for restless bandits with a constraint on the action, rather than on the state. It is not straightforward to the reviewer if the asymptotic proof of Whittle index policy can be generalized to that for WIMS. Third, as discussed above, there is a gap between the problem formulation (with state constraint) and the proposed WIMS (ignore the constraint). To this end, this paper tunes the parameters of $\lambda_g$. Though empirical results are provided, it is unclear if this is theoretically sound given that there is not performance guarantee in this paper.

3. From the experimental results, the proposed WIMS outperformed the considered baselines in terms of accumulated profit. What about the computational complexity? The reviewer did not fully understand the results presented in Table 4. It seems that WIMS takes significant larger computational costs, measured in mins. Given these large numbers (mins), how could this algorithm adapt to dynamic settings?

**Questions:**

See comments in weakness.

---

> ### Author Response · Authors · 2023-11-20
> **Author Response to Reviwer xHVA (1/2)**
>
> Dear Reviewer,
>
> We greatly appreciate your insightful comments and the time you took to review our manuscript. Your feedback is essential to improving our work, and here we address your concerns:
>
> **In this paper, the authors assume that the multiple SKUs are independent and their total inventory level cannot exceed some capacity constraint. Unfortunately, this assumption is problematic and lack of justification. For example, rewards are not only independent in large-scale systems (a large number of SKU), and the substitution effect can occur.**
>
> First, this independence assumption is widely adopted in recent studies on data-driven approaches for inventory management (e.g.,  [Perez et al 2021; Qi et al 2023]). People believe that the rewards/dynamics for different SKUs are largely independent in practice. As for the practical scenario where rewards/dynamics are not fully independent, we believe our solution provides a good approximate result.
>
> Second, this independence assumption plays an important role in making the inventory management problem for large-scale systems tractable. For example, [Perez et al 2021; Qi et al 2023] can hardly account for the interdependency of different SKUs (such as the substitution effect) since modeling such interdependency calls for a simulator or hand-crafted rules that can predict counterfactual scenarios. It is still an open problem to deal with interdependency within a large-scale system.
>
> **First, this paper introduces a state constraint that must be satisfied in each time step in Section 4.1. However, WIMS is NOT guarantee to satisfy this constraint at all, and hence there is a gap between the formulated problem and the proposed solution.**
>
> Although our algorithm cannot guarantee the state constraint always holds, our empirical results show that this constraint strictly holds in most of the cases for our algorithm (see Table 1), and our algorithm incurs much less  constraint violations than existing baselines From the mathematical modeling perspective, it is hard to maintain the constraint strictly when making decisions since we consider a constraint in the state space. From a practical perspective, it is unlikely to strictly prevent constraint violation (exceeding the capacity limit) w/o reducing the profitability. Specifically, there are many practical workarounds when the inventory overflows, such as clearance sale, renting additional storage space, or returning products, and these workarounds are preferable than using an overly conservative and low-profitability strategy.
>
> **Second, it is not quite clear if the proposed WIMS is asymptotically optimal or not, which is a key performance in restless bandits literature. For example, the state-of-the-art Whittle index policy is provably asymptotically optimal. However, Whittle index policy is designed for restless bandits with a constraint on the action, rather than on the state. It is not straightforward to the reviewer if the asymptotic proof of Whittle index policy can be generalized to that for WIMS.**
>
> The state-of-the-art performance guarantee for the Whittle index policy is studied when the number of arms is infinite. For our case, we can prove the optimality of the WIMS policy when the capacity limit is infinite, i.e., $L_t \to \infty$. The proof is as follows: When $L_t \to \infty$, we want to show that the WIMS policy with the global cost $\lambda_g = 0$ is optimal. This policy can be written as
>
> $$ a_t^i = \sum_{\alpha < A} \mathbb{I} [  [f(s_t^i)]_\alpha \ge 0 ] + 1 $$
>
> $$ = \sum_{\alpha < A} \mathbb{I} [  Q_0^*(s, >\alpha) \ge Q_0^*(s, \le \alpha) ] + 1 $$
>
> $$ = arg \max_\alpha Q_0^*(s, \alpha) $$
>
> where the first equation follows the definition of the WIMS policy (Eq. 3), the second equation follows Lemma 4.5, the third equation can be derived following the definition in Eq. 2. Notice that when the capacity limit is infinite, the optimal policy should always take the action $arg \max_\alpha Q_{\lambda = 0}^*(s, \alpha)$.
>
> **Third, as discussed above, there is a gap between the problem formulation (with state constraint) and the proposed WIMS (ignore the constraint). To this end, this paper tunes the parameters of $\lambda_g$. Though empirical results are provided, it is unclear if this is theoretically sound given that there is not performance guarantee in this paper.**
>
> We acknowledge that the gap between the hard constraint in the formulation and our algorithm where the constraint is not strictly satisfied. As explained in the above response, since the transition dynamics can be arbitrary, it is hard to ensure such a state-space constraint is strictly satisfied without significantly sacrificing the profitability. Therefore, we emphasize that we should focus more on the empirical results which show that our algorithm results in a strategy that satisfies the constraint reasonably well (compared to existing baselines).

---

> > ### Comment · Reviewer_xHVA · 2023-11-22
> > **Thank you for your response**
> >
> > Thank you for your response. Appreciate it.
> >
> > 1.  I cannot agree with your argument that "it is hard to maintain the constraint strictly". Note that in the conventional restless multi-armed bandits (RMAB) formulation, there is a natural "hard or instantaneous constraint" that **must** be satisfied at each time step or decision epoch, e.g., at most $K$ arms can be pulled or activated at each time step or decision epoch. In order to satisfy this constraint, Whittle is the first to propose the index policy so as to associate an index value with each arm. Whittle index policy then simply ranks all arms based on their indices at each time step, and activates the highest K ranked arms.  To this end, the "hard or instantaneous constraint" can be exactly satisfied. This is one nature of RMAB problem, and one reason to develop index policy, starting from the seminal Whittle index policy and many others.
> >
> > In case the constraints to be violated, there is a related problem on constrained MDP (CMDP), which received a lot of attentions in recent years. Note that CMDP is different from RMAB in nature, and there are various ways to defined the constraints in CMDP. There are recent works on both model-based or model-free RL methods, to provided either bounded constraint violation regret or without constraint violation. I would suggest the authors to carefully think about what is the nature of your problems, RMAB or CMDP or ?
> >
> > 2. Note that Whittle (1988) proposed the heuristic Whittle index policy and conjectured that it is asymptotically optimal. Weber and Weiss (1990) is the first to provide a rigorous proof that Whittle index policy is asymptotically optimal. The proof is under the asymptotic regime that both the number of arms $N$ and the constraint $K$ must be go to infinitely with their ratio fixed, i.e., $N\rightarrow \infty$, $K\rightarrow \infty$ and $K/N$ fixed (What you described is not precise or correct). In addition, the proofs can go through under a "global attractor" condition. This is a technical assumption that must be needed in most of asymptotic proofs for index policies for RMABs (though there are recent efforts to relax this technical assumption). To this end, the asymptotic proof is not that easy as you described just in one line. Also following the above comment, it is not clear about the problem formulation, not clear how to define or whether the global attractor assumption satisfied for your problem. It is hard for me to understand how to prove the asymptotic optimality of your WIMS policy. If that is true, I would strongly suggest the authors to provide a clear statement, with a rigorous proof in the future version of this paper so that the readers can have an opportunity to take a look.
> >
> > Given these technical issues in this paper, I will keep my original score. In addition, I would strongly suggest the authors seriously consider the comments raised by all reviewers, and if possible, address them to improve your paper before considering a resubmission again.

---

> > > ### Author Response · Authors · 2023-11-23
> > > **Thanks for the Feedback**
> > >
> > > Dear Reviewer xHVA,
> > >
> > > Thanks for your further feedback!
> > >
> > > For the first point, we want to clarify that our problem is not formulated under the RMAB framework and we do not impose a hard constraint on the inventory capacity. We follow a practical formulation for the real-world application where the agents try to minimize the constraint violation without significantly sacrificing profit. Our algorithm is motivated by previous work on RMAB/the Whittle index, but we aim at developing a scalable and adaptive algorithm for real-world usage instead of sticking to an existing framework for theoretical analysis.
> > >
> > > For the second point, since we analyze under the limit $L_t \to \infty$ (indicating that no constraint exists), the proof does not need to involve technical assumptions such as the "global attractor" condition.
> > >
> > > Finally, we sincerely hope the reviewer can re-evaluate our paper from a **practical** perspective considering its **adaptability** (to different constraint levels and number of SKUs, and sudden changes to the capacity limit), which is a new feature that did not exist for previous inventory management algorithms.

---

> ### Author Response · Authors · 2023-11-20
> **Author Response to Reviwer xHVA (2/2)**
>
> **What about the computational complexity? The reviewer did not fully understand the results presented in Table 4. It seems that WIMS takes significant larger computational costs, measured in mins. Given these large numbers (mins), how could this algorithm adapt to dynamic settings?**
>
> One of the main advantages of WIMSN over the previous algorithm is that the model trained by WIMSN is applicable to different scenarios (with different number of SKUs and different capacity limits). Consequently, WIMSN only needs to train the model once and this model can result in good performance in all the scenarios (as shown in Table 1). In contrast, previous algorithms need to train separate models for different scenarios. Let us take the computational costs (measured in the training time on a same machine) of different algorithms shown in Table 4 as an example. If we aim to manage the inventory for 2k SKUs with the four possible capacity levels, we need to maintain 4 IPPO models which cost 1567+1618+1593+1548=6326 minutes to train but the WIMSN-d model only needs 2652 minutes and it outperforms the 4 IPPO models in their trained scenarios.
>
> At last, we want to emphasize again that our paper aims to look for a good learning policy in practice. It would be really appreciated if the reviewer could evaluate our paper more based on the empirical performance.
>
> **References**
> * [Perez et al 2021] Perez, Hector D., et al. "Algorithmic approaches to inventory management optimization." Processes 9.1 (2021): 102.
> * [Qi et al 2023] Qi, Meng, et al. "A Practical End-to-End Inventory Management Model with Deep Learning." Management Science 69.2 (2023): 759-773.

---

> ### Author Response · Authors · 2023-11-22
> **Thank You for the Feedback**
>
> Dear Reviewer xHVA,
>
> As the author reviewer discussion period is ending soon, we sincerely thank you for the invaluable feedback. Should our response effectively address your concerns, we kindly hope that you could consider raising the score rating for our work. We will also be happy to address any additional queries or points.
>
> Best regards,
>
> Authors

---

### Official Review · Reviewer_spvh · 2023-10-31

**Soundness:** 3 good
**Presentation:** 4 excellent
**Contribution:** 3 good
**Rating:** 8
**Confidence:** 3

**Summary:**

This paper extended Whittle index method to a multi-agent reinforcement learning setting under the inventory management setup with multiple discrete actions (number of replenishments) and a global constraint on the state space (total inventory not exceeds a certain limit). The paper bridges two gaps to the restless bandits problem: the constraint is imposed on the state instead of the actions; there are multiple actions for each agent instead of binary actions; by measuring the cost of unit budget consumption and generates the critical points for different actions (changes index into vectors). Then the author proposed a algorithm combines their WIMS with a neural network to solve problem more efficiently. The real data experiments shows that the new policy performs good and efficiently with less constraint violations compared to existing policies.

**Strengths:**

-	This paper introduces a novel adaptation of the Whittle Index tailored for a MARL setting. The adaptation addresses challenges that previously hindered the direct application of the Whittle Index such as multiple actions for each agent and constraints on state spaces instead of action spaces. The application of WIMS in a deep MARL algorithm, called WIMSN, further establishes its originality.
-	This paper compares the proposed algorithm with both traditional operation research methods and other MARL baselines, providing a comprehensive evaluation. The adaptive nature of WIMSN to changes in constraints or combinations of SKUs without requiring retraining emphasizes the quality and flexibility of the proposed approach.
-	This paper is well-structured and organized. Technical concepts such as WIMS are well explained using simple examples, making them easily accessible to readers.
-	The ability to scale WIMSN to thousands of agents highlight its significant, especially in large-scale industrial scenarios.
-	This paper is original in that it measures the cost of unit budget consumption and generates the whittle index to be a vector. Furthermore, the author gave a reasonable sufficient condition of indexability for the vector whittle index, (which is the optimal policy should not reduce the replenishment quantity when the inventory cost decreases.)
-	This paper has high contributions since the methods generally applied to other problems where the global constraint of a multi-agent learning problem depends on states instead of on actions.

**Weaknesses:**

-	This paper builds upon the Whittle index, but there is no comprehensive exploration of the inherent limitations or challenges of using this index in a MARL context. This oversight could result in practical challenges or unintended outcomes when transitioning to real-world implementations. It would be beneficial for the authors to outline specific assumptions made when integrating the Whittle Index. One potential area of exploration could be the interplay of local constraints in conjunction with global constraints. Some SKUs might have their own unique storage.
-	This paper does not have details on how the dataset is partitioned into training, validation, and testing subsets. Providing this information is essential for ensuring replicability and comprehending the robustness of the outcomes. It would be beneficial for the authors to explicitly define and justify their splitting methodology, particularly given the time series nature of the data.
-	It would be helpful if the authors mentioned areas where their method (WIMSN) could be improved or further developed. By discussing potential future studies or enhancements, the authors would make their paper even more valuable.
-	The literature review seems not very complete and up-to-date. The author reviewed a lot of works on the operation research and on independent learning based MARL without global constraints, and the main comparisons were made with works in 217,2018. I am wondering if there are more recent works focus on similar topic.
-	One of the focuses, main assumptions and challenges in this paper, which is the limit of the total inventory capacity changes across the different seasons is not very reasonable to me. If the constraint is only changed once a long time like once a seasons, it seems to me that wrong those previous algorithms and re-train the model does not bring many troubles.

**Questions:**

-	Based on the above mentioned weaknesses, I feel that the comparison in Figure 2 does not complete or fair enough.  The author compared their adaptive algorithm with two IPPO models trained with two different capacity limits, but the author didn’t mention any training cost or training time for IPPO, so I am wondering why the IPPO users can’t re-train their models on the 50th day, since the reward was compared on a daily manner.
-	In the neural network training phase, in the network update steps, they sample a batch of transitions from the replay buffer, could this word “batch” be more clarified? And is this replay buffer something important to computational efficient of the algorithm? What is the difference from regular learning algorithms that runs a sequence of data first and without a replay buffer?
-	OR-based methods was first mentioned without explanation, should add an abbreviation after first mentioning operation research.
-	It's unclear from the given text what dataset was used for training and testing the proposed methods. This raises concerns about the generalizability and robustness of the proposed models. Are the datasets representative of real-world scenarios? Are they publicly available for verification and reproducibility?
-	While the paper mentions computational costs and standard deviations of daily profits, are there other metrics (like fill rate, stockout rate, etc.) used in inventory management that could provide a more comprehensive view of performance?
-	(Typo in Proposition 3.3) In the third case, Q(s,1) = Q(s,i) -> Q(s,1) = Q(s,0)

---

> ### Author Response · Authors · 2023-11-20
> **Author Response to Reviwer spvh (1/3)**
>
> Dear Reviewer,
>
> We greatly appreciate your insightful comments and the time you took to review our manuscript. Your feedback is essential to improving our work, and here we address your concerns:
>
> **There is no comprehensive exploration of the inherent limitations or challenges of using this index in a MARL context. It would be beneficial for the authors to outline specific assumptions made when integrating the Whittle Index.**
>
> In the MARL context considered in our paper, we integrate the Whittle index as a tool to decrease the effective size of the joint action space of multiple agents. The integration of Whittle index to the MARL context relies on the **weak coupling** problem structure mentioned in the paper, i.e., one possibly changing global constraint couples the otherwise independent dynamics of multiple agents. In other words, the independence of different agents (except the only global constraint) is the key assumption when integrating WIMS to the MARL setting.
>
> **This paper does not have details on how the dataset is partitioned into training, validation, and testing subsets. It's unclear from the given text what dataset was used for training and testing the proposed methods. This raises concerns about the generalizability and robustness of the proposed models. Are the datasets representative of real-world scenarios? Are they publicly available for verification and reproducibility?**
>
> Thanks for mentioning the dataset and we will add this information in the appendix. This dataset is provided by our partner who is a world-wide retail corporation, and thus we believe the dataset is representative of real scenarios. We will try to make the dataset public available after de-identification.
>
> The experiments in our paper is based on the history of 2307 SKUs from 2018-08-01 to 2019-12-13. The training / validation / testing sets are set to be the subset 2018-08-01 to 2019-05-27 (300 days) / 2019-05-28 to 2019-09-04 (100 days) / 2019-09-05 to 2019-12-13 (100 days) respectively. For the experiments on 100/1000 SKUs, we sample a subset of SKUs uniformly randomly from the 2307 SKUs.
>
> **It would be helpful if the authors mentioned areas where their method (WIMSN) could be improved or further developed. By discussing potential future studies or enhancements, the authors would make their paper even more valuable.**
>
> Thanks for the good suggestion. The possible future directions are as follows: 1) Application: In principle, WIMS can solve a wider range of tasks as long as the **weak coupling** structure exists. For example, in portfolio management for quantitative investment, the investment on different stocks is largely independent but subject to a global constraint on the total asset. 2) Theory: Although the Whittle index usually achieves good performance in practice, it lacks performance guarantee in theory. We may try to achieve theoretical guarantee for applying the Whittle index (or its variance such as WIMS) to the MARL setting - possibly for some asymptotic case (such as the new asymptotic result where the optimality of the WIMS policy when the capacity limit is infinite, i.e., $L_t \to \infty$, is shown; see the response to Q1 of Reviewer qxXR for the proof).
>
> **The author reviewed a lot of works on the operation research and on independent learning based MARL without global constraints, and the main comparisons were made with works in 217,2018.**
>
> In the MARL field, researchers also focus on value decomposition based methods (such as VDN [Sunehag et al 2017], QMIX [Rashid et al 2020], QTRAN [Son et al 2019], and HARL [Zhong et al 2023]) and policy gradient based methods (such as MADDPG [Lowe et al 2017], COMA [Foerster et al 2018], and MAPPO [Yu et al 2022]). However, these algorithms require unaffordable computational resources in our case (with 2k+ agents). Therefore, we only compare our algorithms with independent learning based MARL algorithms in our experiments.

---

> ### Author Response · Authors · 2023-11-20
> **Author Response to Reviwer spvh (2/3)**
>
> **If the constraint is only changed once a long time like once a seasons, it seems to me that wrong those previous algorithms and re-train the model does not bring many troubles. I feel that the comparison in Figure 2 does not complete or fair enough. The author compared their adaptive algorithm with two IPPO models trained with two different capacity limits, but the author didn’t mention any training cost or training time for IPPO, so I am wondering why the IPPO users can’t re-train their models on the 50th day, since the reward was compared on a daily manner.**
>
> Our algorithm is advantageous over the previous algorithms that need re-training in the following aspects: 1) Our algorithm decouples the training phase and the deployment phase, which means that the end user of the model (e.g., the store manager accessing limited or no computational resources in practice) do not need to be involved in model training. 2) Our algorithm is adaptive to not only the changing capacity limit but also the number of SKUs, whereas previous algorithms are not adaptive to the changing capacity limit nor the number of SKUs. Adding new SKUs or delisting SKUs occurs frequently in practice. As for the comparison on the training time, the training time of our algorithms is less than 2x the training time of previous algorithms for the same number of SKUs. See details in Table 4 in the appendix.
>
> **In the neural network training phase, in the network update steps, they sample a batch of transitions from the replay buffer, could this word “batch” be more clarified? And is this replay buffer something important to computational efficient of the algorithm? What is the difference from regular learning algorithms that runs a sequence of data first and without a replay buffer?**
>
> In Phase 1 of our algorithm, we largely follow the framework of DQN [Minh et al 2015] given the similarity of Eq (4) to the TD update in DQN. Accordingly, we adopt a similar design for the replay buffer. Specifically, the capacity of the replay buffer is set to 10M and the oldest samples are removed from the buffer if it is full. For each time, we sample a batch of 16K samples uniformly randomly from the replay buffer and use these samples to train the Q-network and the Whittle index network. This naive replay buffer does not influence computational efficiency significantly, in contrast to its variants such as prioritized experience replay [Schaul et al 2015]. We adopt the replay buffer to prevent the instability issue for such off-policy RL algorithms.

---

> ### Author Response · Authors · 2023-11-20
> **Author Response to Reviwer spvh (3/3)**
>
> **OR-based methods was first mentioned without explanation, should add an abbreviation after first mentioning operation research.**
>
> Thanks for pointing this out. We will revise this.
>
> **While the paper mentions computational costs and standard deviations of daily profits, are there other metrics (like fill rate, stockout rate, etc.) used in inventory management that could provide a more comprehensive view of performance?**
>
> Thanks for the good suggestion. We will provide more metrics for evaluation in the appendix. In terms of fill rate and stockout rate, we observe that they are largely correlated with the profit metric presented in Table 1.
>
> **Typo in Proposition 3.3.**
>
> Thanks for pointing it out and we will revise this.
>
> **References**
>
> * [Sunehag et al 2017] Sunehag, Peter, et al. "Value-decomposition networks for cooperative multi-agent learning." arXiv preprint arXiv:1706.05296 (2017).
> * [Rashid et al 2020] Rashid, Tabish, et al. "Monotonic value function factorisation for deep multi-agent reinforcement learning." The Journal of Machine Learning Research 21.1 (2020): 7234-7284.
> * [Son et al 2019] Son, Kyunghwan, et al. "Qtran: Learning to factorize with transformation for cooperative multi-agent reinforcement learning." International conference on machine learning. PMLR, 2019.
> * [Lowe et al 2017] Lowe, Ryan, et al. "Multi-agent actor-critic for mixed cooperative-competitive environments." Advances in neural information processing systems 30 (2017).
> * [Foerster et al 2018] Foerster, Jakob, et al. "Counterfactual multi-agent policy gradients." Proceedings of the AAAI conference on artificial intelligence. Vol. 32. No. 1. 2018.
> * [Yu et al 2022] Yu, Chao, et al. "The surprising effectiveness of ppo in cooperative multi-agent games." Advances in Neural Information Processing Systems 35 (2022): 24611-24624.
> * [Zhong et al 2023] Zhong, Yifan, et al. "Heterogeneous-Agent Reinforcement Learning." arXiv preprint arXiv:2304.09870 (2023).
> * [Minh et al 2015] Mnih, Volodymyr, et al. "Human-level control through deep reinforcement learning." nature 518.7540 (2015): 529-533.
> * [Schaul et al 2015] Schaul, Tom, et al. "Prioritized experience replay." arXiv preprint arXiv:1511.05952 (2015).

---

### Meta-Review · Area_Chair_rYE5 · 2023-12-10

**Metareview:**

This paper extends Whittle index method to a multi-agent reinforcement learning setting under the inventory management setup with multiple discrete actions (number of replenishments) and a global constraint on the state space (total inventory not exceeds a certain limit). Such algorithmic extension is novel and extensive empirical results are provided to demonstrate its efficacy. That said, the theoretical aspect of the paper is a bit unsatisfying. Asymptotic optimality in constrained case and any finite-sample results are currently lacking. That said, my reading is that this can still be an interesting, non-trivial first step in the multi-agent inventory management problem, and its publication would stimulate further research in this area.

**Justification For Why Not Higher Score:**

Lack of finite-sample bounds.

**Justification For Why Not Lower Score:**

NA

---

### Decision · Program_Chairs · 2024-01-16

Accept (poster)